



Title: **Agropedogenesis: Humankind as the 6th soil-forming factor and attractors of agrogenic soil degradation**

Authors: Yakov Kuzyakov[1,2], Kazem Zamanian[1*]

[1.] Department of Soil Science of Temperate Ecosystems, Georg-August University of Göttingen, Büsgenweg 2, 37077 Göttingen, Germany

[2.] Department of Agricultural Soil Science, Georg-August University of Göttingen, Büsgenweg 2, 37077 Göttingen, Germany

* Author for correspondence

Kazem Zamanian

Phone: +49 (0)551 39 12104

E-mail: zamanians@yahoo.com

Keywords: Anthropogenic soil change, Soil formation and degradation, Soil forming factors, Pedogenesis, Agropedogenesis, Land use, Intensive agriculture, Soil erosion, Anthropocene





**Abstract**

Agricultural land covers 5100 million ha (ca. 50% of potentially suitable land area) and agriculture has immense effects on soil formation and degradation. Although, the concepts or theories of agropedogenesis have already been advanced; but still need further consideration to better understand the dynamics of soil development under agricultural practices. We introduce a theory of *anthropedogenesis* – soil development under the main factor 'humankind' – the 6[th] factor of soil formation, and deepen it to encompass *agropedogenesis* as the most important direction of anthropedogenesis. The developed theory of agropedogenesis consists of (1) broadening the classical concept of Factors – Processes – Properties with the addition of Functions along with their feedbacks to the Processes, (2) a new concept of attractors of soil degradation, (3) selection and analysis of master soil properties, (4) analysis of phase diagrams of master soil properties to identify thresholds and stages of soil degradation, and finally (5) a definition of multi-dimensional attractor space of agropedogenesis. We show that the factor 'humankind' dominates over the effects of the five natural soil-forming factors and that agropedogenesis is therefore much faster than natural soil formation. The direction of agropedogenesis is mainly opposite to that of natural soil development and is thus mainly associated with soil degradation. In contrast to natural pedogenesis leading to divergence of soil properties, agropedogenesis leads to their convergence because of the efforts to optimize conditions for crop production. Agricultural practices lead soil development toward a quasi-steady state with a predefined range of measured properties – attractors (an attractor is a minimal or maximal value of a soil property, toward which the property will develop via long-term intensive agricultural use from any natural state). Based on phase diagrams and expert knowledge, we define a set of 'master properties' (bulk density and macroaggregates, soil organic matter content and pH, microbial biomass and basal respiration). These master properties are especially sensitive to land use and determine the other properties during agropedogenesis. Phase diagrams of master soil properties help identify thresholds and stages of soil degradation. Combining individual attractors to a multi-dimensional attractor space enables predicting the trajectory and the final state of agrogenic soil development and to develop measures to combat soil degradation.

*Keywords*: Anthropogenic soil change, Soil formation and degradation, Soil forming factors, Pedogenesis, Agropedogenesis, Land use, Intensive agriculture, Soil erosion, Anthropocene

## 1. Introduction

### 1.1. Soil degradation by agricultural land-use

Soils (S) as natural bodies are formed via interactions of soil-forming factors, i.e. climate (cl), organisms (o), relief (r), and parent material (p) over time (t) (Dokuchaev, 1883; Glinka, 1927; Jenny, 1941): S = f(cl, o, r, p, t, ...).

The processes of additions, losses, transfers/translocation, and transformations of matter and energy over centuries and millennia produce a medium – soil (Simonson, 1959), which supports plant roots and fulfils many other ecosystem functions (Lal, 2008; Nannipieri et al., 2003; Paul, 2014). These functions however, commonly decrease due to human activities, in particular through agricultural practices because of accelerating soil erosion, nutrient loss (despite intensive fertilization), aggregate destruction, compaction, acidification, alkalization and salinization (Homburg and Sandor, 2011;



Sandor and Homburg, 2017). Accordingly, the factor 'humankind' has nearly always been considered as a soil-degrading
entity that, by converting natural forests and grasslands to arable lands, changes the natural cycles of energy and matter.
Except rare cases which are leading to the formation of fertile soils such as *terra preta* in the Amazonian Basin (Glaser et
al., 2001), *plaggen* in North Europe (Pape, 1970) as well as *hortisols* (Burghardt et al., 2018), soil degradation is in most
cases the outcome of long-term agricultural practices (DeLong et al., 2015; Homburg and Sandor, 2011). Soil degradation
begins immediately after conversion of natural soil coverage and land preparation for cultivation and involves the
degradation in all physical, chemical and biological properties (Table 1). The result is a decline in ecosystem functions.
This degradation gains importance when considering the rapid increase in human populations (Carozza et al., 2007)
and technological progress. Increasing food demand necessitates either ever larger areas for croplands or/and
intensification of crop production per area of already cultivated land. Since the suitable land resources for agriculture are
limited and increasingly located in ecologically marginal conditions, any increase in food production will depend on the
second option: intensification. This will intensify the imbalance between input to and output from the soil, resulting in
faster and stronger soil degradation. While prohibiting or reducing degradation is essential in achieving sustainable food
production (Lal, 2009), many studies have addressed individual mechanisms and specific drivers of soil degradation
(Table 1). Nonetheless, there is still no standard and comprehensive measure to determine soil degradation intensity and
to differentiate between degradation stages.
Agricultural soils (croplands + grasslands) cover 5100 million ha, corresponding to about 34% of the global land
area. Importantly, huge areas are located in very cold regions that are continuously covered by ice (1500 million ha),
located in hot deserts, mountainous areas, or barren regions (2800 million ha), as well as sealed in urban and industrial
regions and roads (150 million ha). Accordingly, agricultural lands cover about 50% of the area potentially suitable for
agriculture (*https://ourworldindata.org/yields-and-land-use-in-agriculture*). Even though huge areas of land are occupied
by agriculture, and humans have modified natural soils over the last 10-12 thousand years, the theory of soil formation as
affected by humankind – anthropedogenesis and its subcategory agropedogenesis – is still far from proper attention. This
paper therefore presents for the first time a theory of *anthropedogenesis* – soil development under the main factor
'humankind' – the 6[th] factor of soil formation. Moreover, we expand it to encompass *agropedogenesis* as a key aspect of
general anthropedogenesis.

**1.2.  Humans as the main soil-forming factor**
Humans began to modify natural soils with the onset of agriculture ca. 10-12 thousand years ago (Diamond, 2002),
resulting in soil degradation. Examples of soil degradation leading to civilization collapses are well known starting at
least from Mesopotamia (18[th] to 6[th] centuries BC) (Diamond, 2002; Weiss et al., 1993). Notwithstanding all negative
impacts of human on soils and on cycles of energy and matter, the intention was always to increase fertility to boost crop
production (Richter et al., 2011; Sandor and Homburg, 2017), reduce negative environmental consequences, and achieve
more stable agroecosystems. To attain these aims, humans have (i) modified soil physical and hydrological properties (for
example, by removing stones, loosening soil by tillage, run-off irrigation, terracing), (ii) altered soil chemical conditions
through fertilization, liming, desalinization, and (iii) controlled soil biodiversity by sowing domesticated plant species
and applying biocides (Richter et al., 2015). Although these manipulations commonly lead to soil degradation (Homburg



and Sandor, 2011; Paz-González et al., 2000; Sandor et al., 2008), they are aimed at decreasing the most limiting factors
(nutrient contents, soil acidity, water scarcity, etc.) for crop production, regardless of original environmental conditions in
which the soil was formed (Guillaume et al., 2016a; Liu et al., 2009). Thus, agricultural land-use always focused on
removing limiting factors and providing optimal growth conditions for a few selected crops: 15 species make up 90% of
the world's food, and 3 of them – wheat, corn, and rice – supply 2/3 of this amount. These crops have similar water and
nutrient requirements (except rice) compared to the plants growing under natural conditions. Consequently, agricultural
land-use has always striven to narrow soil property space to uniform environmental conditions. Examples include long-
term increases in soil moisture via irrigation in arid regions to change the effects of climate (Asperen et al., 2014; Boix-
Fayos et al., 2001; Delgado et al., 2007; Homburg and Sandor, 2011); the application of mineral and organic fertilizers to
overcome the nutrient limitations of parent materials (Liu et al., 2009) and low N fixation; liming the soil to the optimal
pH range for crops.

The human factor can even change soil types as defined by classification systems (Supplementary Fig. 1) by inducing

erosion, changing the thickness of horizons and their mixture, decreasing soil organic matter (SOM) content, destroying
aggregates, and accumulating salts (Dazzi and Monteleone, 2007; Ellis and Newsome, 1991; Shpedt et al., 2017). A
Mollisol (~ Chernozems or Phaeozems), for example, turns into an Inceptisol (~ Cambisols) by decreasing total SOM (Lo
Papa et al., 2013; Tugel et al., 2005) or/and thinning of the mollic epipedon by tillage and erosion and destroying granular
and sub-polyedric structure (Ayoubi et al., 2012; Lo Papa et al., 2013). Accordingly, humankind can no longer be treated
as only a soil-degrading but also as a soil-forming factor (Amundson and Jenny, 1991; Dudal, 2004; Richter et al., 2015;
Sandor et al., 2005). The result is the formation of anthropogenic soils (soils formed under the main factor 'humankind').
This is very well known for rice paddies, i.e. Hydragric Anthrosols (Chen et al., 2011; Cheng et al., 2009; Kölbl et al.,
2014; Sedov et al., 2007) as well as Hortic Anthrosols (long-term fertilized soils with household wastes and manure) and
Irragric Anthrosols (long-term irrigated soils in dry regions) (WRB, 2014). These effects have stimulated the on-going
development of soil classifications to reflect new directions of soil evolution: *anthropedogenesis*, i.e. soil genesis under
the main factor 'humankind' and in particular *agropedogenesis*, i.e. soil genesis under agricultural practices as a
subcategory of anthropedogenesis (Bryant and Galbraith, 2003).

Human impacts on soil formation immensely accelerated in the last 50-100 years (Dudal, 2004) with the (1)

introduction of heavy machinery, (2) application of high rates of mineral fertilizers, especially after discovery of N
fixation by the Haber-Bosch technology, (3) application of chemical plant protection, and (4) introduction of crops with
higher yield and reduced root systems. We expect that despite various ecological measures (no-till practices, restrictions
of chemical fertilizer applications and heavy machinery, etc.) the effects of humans on soil formation will increase in the
Anthropocene and will be even stronger than for most other components of global change. This urgently calls for a
concept and theory of soil formation under humans as the main factor.

**2.    Concept of Agropedogenesis**

*Anthropedogenesis* is the soil formation under the main factor 'humans' (Amundson and Jenny, 1991; Bidwell and

Hole, 1965; Howard, 2017; Meuser, 2010; Yaalon and Yaron, 1966). *Agropedogenesis* is the dominant form of
anthropedogenesis and includes soil formation under agricultural use – mainly cropland (Sandor et al., 2005). The other
forms of anthropedogenesis are construction of completely new soils (Technosols e.g. Urban soils or Mine soils). These





other forms of anthropedogenesis will not be described in this paper, because they are not directly connected with
agriculture.

Agropedogenesis should be clearly separated from the natural pedogenesis because of: (1) strong dominance of the
factor 'humans' over all other five factors of soil formation, (2) new processes and mechanisms that are not present under
natural soil development (Table 2), (3) new directions of soil developments, compared to natural processes (Table 2), (4)
frequent development of processes in the reverse direction compared to natural pedogenesis, (5) much higher intensity of
many specific processes compared to natural developments and consequently faster rates of all changes.

Agropedogenesis and natural pedogenesis are partly opposite processes. Natural soil formation involves the
development of soils from parent materials under the effects of climate, relief, organisms and time. Here, soil formation
will reach quasi-steady state conditions typical for the combination of the five soil-forming factors (Fig. 1).
Agropedogenesis, in most cases, is a process of losing soil fertility i.e. degradation because of intensive agriculture and
narrowing of soil properties. Agropedogenesis is partly the reverse of soil formation but the final stage is not the parent
material (except on a few cases of extreme erosion). Agropedogenesis also leads to a quasi-steady state of soils (Fig. 1)
(Eleftheriadis et al., 2018; Wei et al., 2014). The time needed to reach this quasi-steady state, however, is much shorter
(in the range of a few centuries, decades, or even less) than in natural pedogenesis, which involves millennia (Tugel et al.,
2005). The range of soil properties at this quasi-steady state condition will show the end-limit of agricultural effects on
soil development.

Our theory of agropedogenesis is based on: (1) Concept of 'Factors → Processes → Properties → Functions', (2)
Concept of 'attractors of soil degradation', (3) Selection and analysis of 'master soil properties', (4) Analysis of phase
diagrams between the 'master soil properties' and identification of thresholds and stages of soil degradation, and (5)
'Multi-dimensional attractor space'.

**2.1.  Concept: Factors, Processes, Properties and Functions**

The original concept of "Soil Factors → Soil Properties" (Dokuchaev, 1883; Jenny, 1941) was modified by
"processes", which depend on the factors of soil formation and develops the properties (Gerasimov, 1984). This triad
enables understanding the development of soils from initial parent materials by the effects of climate, relief, vegetation
and organisms over time. Thus, morphological soil properties that are visible in the field and measurable in the lab are
very well described and yielded various (semi)genetic soil classifications (KA-5, 2005; KDPR, 2004; WRB, 2014).

Considering the recent development of functional approaches and ecosystem perspectives, this triad is insufficient.
We therefore introduce the concept: "Factors → Processes → Properties → Functions" (Fig. 3). We do not describe here
the very broad range of functions of natural soils as related to clean air and water, biodiversity, decontamination of
pollutants, biofuel and waste management, etc., but refer to excellent reviews focused on soil functions (Lal, 2008;
Nannipieri et al., 2003).

One function – production – is, however, crucial for agropedogenesis (Fig. 2); because humans change, adapt and
modify natural soils to maximize crop yields. As it is not possible to simultaneously maximize all functions, the functions
other than 'production' decrease or even disappear. Accordingly, *agropedogenesis is driven by processes pursuing the*
*maximization of only one function – crop production*. The consequence is that all other soil functions are reduced. *We*
*define soil degradation as a reduction of functions*. Initially, all functions will be reduced at the cost of increased crop

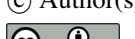


production. As degradation advances, however, the production function decreases as well. Nearly all previous definitions of soil degradation were based on declining crop productivity. The principal difference between our concept of soil degradation and the most common other concepts is that the degradation starts with the reduction of one or more functions – before crop productivity decreases. This concept, based on multi-functionality, is much broader and considers the ecosystem functions and services of soil and the growing human demand for a healthy environment.

Agropedogenesis clearly shows that the natural sequence 'Factors → Processes → Properties → Functions' is changed by humans: Functions are no longer the final step in this sequence because *the functions become a factor* (Fig. 2). This is because humans tailor the processes of soil development for the main function of agricultural soils – productivity. Based on the example of agropedogenesis, we conclude that all types of anthropedogenesis are directed at the functions which humans desire from the soil; hence, the *functions are getting the factors of soil development*.

### 2.2. Attractors of soil degradation: definitions and concept

Despite a very broad range of individual properties of natural soils, long-term intensive agricultural land-use strongly narrows (Homburg and Sandor, 2011; Kozlovskii, 1999; Sandor et al., 2008) their range and ultimately brings individual properties to the so-called attractors of degradation (Kozlovskii, 1999). We define:

**An attractor of a soil property is a numerical value toward which the property tends to develop from a wide variety of initial or intermediate states of pedogenesis.**

**An attractor of agrogenic soil degradation is a minimal or maximal value of a soil property toward which the property tends to develop by long-term intensive agricultural practices from a wide variety of initial conditions common for natural soils.**

Attractors of soil properties are common for natural pedogenesis and anthropedogenesis (Fig. 1). The well-known examples of natural pedogenic attractors are the maximal SOM accumulation (C ≈ 5-6% for mineral soils), highest increase of clay content in the Bt horizon by a ~ two-fold illuviation compared to the upper horizon (without lithological discontinuity), the upper depth of the Bt horizon for sheet erosion, a minimal bulk density of mineral soils of ~ 0.8 g cm³, the maximal weathering in wet tropics by removal of all minerals until only Fe and Al oxides remain (Chadwick and Chorover, 2001).

Natural pedogenesis leads to a divergence of pedogenic properties and consequently to the broadening of the multi-dimensional attractor space (see below) because various soils develop to steady state from the same parent materials depending on climate, relief and organisms (Fig. 1). The time necessary for natural processes to reach these attractors is at least 1-2 orders of magnitude longer than the periods for attractors of agropedogenesis (see below).

In contrast to natural pedogenesis, agropedogenesis narrows the soil properties by optimizing environmental conditions for agricultural crops with similar requirements (Lo Papa et al., 2011, 2013). Consequently, each soil property follows a trajectory from a specific natural level toward the unified agrogenic attractor (Fig. 1). Therefore, in contrast to *Natural pedogenesis resulting in divergence of soil properties*, *Agropedogenesis leads to convergence of soil properties*.




Note that though, convergence is common but may not always hold true as soil behavior and changes are complex with
many causal factors and interactive multiple processes.

**2.3. Examples of attractors of soil degradation**

The convergence in soil properties and thus reaching an attractor after having started from various initial states is
evident by comparing soils under long-term (e.g. centuries) cultivation (Sandor and Homburg, 2017). The challenges that
ancient farmers faced were fundamentally the same as today, albeit with a much stronger intensification of chemical
impacts (fertilization, pesticides) and heavy machinery in the last decades (Dudal, 2004; Sandor and Homburg, 2017).
*The main difference between soil degradation in the past and in the modern era is the rates and extent, but not the*
*processes or mechanisms*. The dynamics of soil properties in long-term cultivations have revealed a narrowing in the
measured values of a given property over time, i.e. a tendency toward the attractor of that property (Alletto and Coquet,
2009; Dalal and Mayer, 1986b; Dalal and J. Mayer, 1986; Haas et al., 1957; Nyberg et al., 2012) (Fig. 3 and 4).
Continuous agricultural practices also decrease the temporal and spatial variability of all properties in the topsoil – in the
Ap horizon (Jones and Dalal, 2017; Scott et al., 1994) (Fig. 5).

In reaching the attractor values, however, the process rates and dynamics differ among various soil properties (Fig.
6), in various geo-climatological regions (Chen et al., 2011, p.29011; Guillaume et al., 2016a; Hartemink, 2006) and
according to land-use intensity. For example, microbial biomass carbon (C) (Henrot and Robertson, 1994) and aggregate
stability (Wei et al., 2014) respond faster than SOM and total N to cultivation. Cultivation affects total N and P content
less than organic C because of N and P fertilization (Guillaume et al., 2016b), whereby a strong decrease of C input is
inferred by the decreasing C:N ratio with cultivation duration (Wei et al., 2014). Whereas cultivation on deforested lands
in the tropics can lead to soil degradation within a few years, converting temperate prairies and steppes to agricultural
fields supports crop production without fertilization for decades (Tiessen et al., 1994). Generally, the degradation rates
(e.g. C losses) in the moist tropics are faster (e.g. about 4-fold) than in the dry tropics (Hall et al., 2013). Despite the
differences in rates, however, the long-term cultivated soils ultimately reach similar degradation levels (Lisetskii et al.,
2015) (Fig. 3f).

**2.4. Master soil properties**

Soils and their functions are characterized by and are dependent on the full range of physical, chemical and
biological properties. A selected few of these properties – the master soil properties – however, are responsible for a very
broad range of functions and define other properties (Lincoln et al., 2014; Lisetskii et al., 2013). *We define a soil property*
*as being a master property if it has a strong effect on a broad range of other properties and if it cannot be easily assessed*
*based on the other properties*. For natural pedogenesis, such master properties – inherited partly from the parent material
– are: clay mineralogy and $CaCO_3$ content, texture, nutrient content, and bulk density. The master properties which are
cumulated or formed during pedogenesis are: soil aggregation/structure, depth of A+B horizons, SOM stock and C:N
ratio, pH, electrical conductivity, etc. (Table 3). These properties largely define the other properties and soil functions
under natural conditions and generally under agricultural use as well.

The master properties of agropedogenesis may differ from those of natural soil development. The crucial difference
is that *the master properties of agropedogenesis must* sensitively respond to agricultural use over the cultivation period.





Accordingly, properties such as texture, clay content and mineralogy – crucial master properties of natural pedogenesis,
are unimportant for agropedogenesis. Note that, although these properties may change under certain circumstances
(Karathanasis and Wells, 1989; Velde and Peck, 2002), they fail to qualify as master properties in agropedogenesis
because they are relatively insensitive to agricultural land-use.
Master soil properties have an additional important function: they are (co)responsible for the changes in other
properties. Changes in a master property over time may therefore intensify or dampen changes in other (secondary)
properties. The stability of macroaggregates, for example, increases with the content and quality of SOM (Boix-Fayos et
al., 2001; Celik, 2005). The infiltration rate and water holding capacity decreases with increasing bulk density (Rasa and
Horn, 2013; Raty et al., 2010), promoting erosion. These relations between soil properties, however, seem to be
significant only within certain ranges, i.e. until thresholds are reached. Beyond such thresholds, new relations or new
master properties may govern. For example, an increasing effect of SOM content on aggregate stability in extremely arid
regions of the Mediterranean was recorded at above 5% SOM contents (Boix-Fayos et al., 2001). Increasing organic
matter contents up to this 5% threshold had no effect on aggregate stability: instead, the carbonate content was the main
regulator (Boix-Fayos et al., 2001). Microbial biomass and respiration in well-drained Acrisoils in Indonesia are resistant
to decreasing SOM down to 2.7% of SOM, but strongly dropped beyond that value (Guillaume et al., 2016b). While the
amounts of SOM and total N in sand and silt fractions may continuously decrease with cultivation duration, those values
in the clay fraction remain stable (Eleftheriadis et al., 2018) (Fig. 3e). Bulk density increases non-linearly with SOM
decrease, and the rates depend on SOM content (Fig. 7). Phase diagrams are very useful to identify such thresholds (see
below).
Summarizing, we define '*Master properties*' as a group of soil-fertility-related parameters that (1) are directly
affected by management – are sensitive to agricultural use and soil degradation, (2) determine the state of many other
(non-master) parameters and soil fertility indicators during agropedogenesis, and (3) should be orthogonal to each other,
i.e. independent (or minimally dependent) of one other (Kozlovskii, 1999), modified). Note that, in reality all soil
properties are at least partly dependent. Nonetheless, the last prerequisite – orthogonality – ensures the best separation of
soils in multi-dimensional space (see below) and reduces the redundancy of the properties.
Considering the three prerequisites and based on expert knowledge, as well as on phase diagrams (see below), we
suggest 8 properties as being master (Table 3): Density; Macroaggregates, SOM, C/N ratio, pH, EC, Microbial biomass
C, and Basal respiration. We consider these 8 to be sufficient to describe the degradation state of most other parameters
during agropedogenesis and to define their multi-dimensional attractor space (see below). Their definition enables
assessing the other properties: water permeability, penetration resistance, erodibility, base saturation, exchangeable
sodium percentage, sodium absorption ratio, N mineralization, availability of other nutrients, etc.
The combination of master properties provides a minimum dataset to determine soil development stages with
cultivation duration (Andrews et al., 2002). Organic C content is the most important and universally accepted master
property that directly and indirectly determines the state of many physical (soil structure, density, porosity, water holding
capacity, percolation rate, erodibility) (Andrews et al., 2003; Nabiollahi et al., 2017; Shpedt et al., 2017), chemical
(nutrient availability, sorption capacity, pH) (Lal, 2006; Minasny and Hartemink, 2011), and biological (biodiversity,
microbial biomass, basal respiration) (Raiesi, 2017) properties. The values of the mentioned secondary properties can be
estimated with an acceptable uncertainty based on robust data on SOM content (Gharahi Ghehi et al., 2012). Finding





additional soil properties beyond SOM to form the set of master properties is, however, not straightforward (Homburg et
al., 2005) because it depends on the desired soil functions (Andrews et al., 2003) such as nutrient availability, water
permeability and holding capacity, crop yield quantity and quality, etc. (Andrews et al., 2002). Therefore, various types of
master properties, depending on geo-climatological conditions (Cannell and Hawes, 1994), have already been suggested
(Table 3). Nonetheless, the dynamics, sensitivity and resistance of such properties to degradation and with cultivation
duration are unknown (Guillaume et al., 2016b).

**2.5. Analysis of phase diagrams and identification of thresholds and stages of soil degradation**

All the properties described above move toward their attractors over the course of soil degradation with time (Fig. 3
and 6). The duration, however, is difficult to compare between soils because the process rates depend on climatic
conditions and land-use intensities. One option to understand and analyze soil degradation independent of time is to use
phase diagrams. Phase diagrams present (and then analyze) properties against each other, without the time factor (Fig. 7c
and 8). Thus, various properties measured in a chronosequence of soil degradation are related to each other on 2D or even
3D graphs (Fig. 9), and time is excluded.

Phase diagrams have two advantages: (1) they help evaluate the dependence of properties on each other – independent
of time, climate, or management intensity. They represent generalized connection between the properties. This greatly
simplifies comparing the trajectory of soil degradation under various climatic conditions, management intensities and
even various land-uses. (2) Such diagrams enable identifying the *thresholds* and stages of soil development and
degradation.
We define:
***Thresholds*** **of soil development and degradation are relatively abrupt changes in process rates or process**
**directions leading to a switch in the dominating mechanism of soil degradation.**

***Stages*** **of soil degradation are periods confined by two thresholds and characterized by one dominating**
**degradation mechanism** (Fig. 7c)**.**

Importantly, soil degradation does not always follow a linear or exponential trajectory (Kozlovskii, 1999). This means
that changes (absolute for linear or relative for exponential) are not proportional to time or management intensity. Soil
degradation proceeds in stages of different duration and intensity. The key consideration, however, is that each stage is
characterized by the dominance of one (group) of degradation processes, whose prerequisite is formed in the previous
phase.

We conclude that phase diagrams (1) enable tracing the trajectory of various soil properties as they reach their
attractors, independent of time, land-use or management intensity, and (2) are useful into analyze not only the dependence
(or at least correlation) between individual properties, but also to identify the thresholds of soil degradation. The
thresholds clearly show that soil degradation proceeds in stages (Fig. 7c, 8 and 9), each of which is characterized by the
dominance of one specific degradation process with its specific rates (and affecting the degradation of related soil
properties).

**2.6. Multi-dimensional attractor space**



The phase diagrams described above were presented in 2D or 3D space and help to evaluate the connections between
the properties and the stages of soil degradation. The suggested 8 master soil properties are orthogonal and the phase
diagrams can therefore be built in multi-dimensional attractor space – the space defining the soil degradation trajectory
based on the master soil properties (Fig. 9 bottom). Therefore, **Development of master soil properties during long-**
**term intensive agricultural land-use and degradation forms a multi-dimensional space of properties (multi-**
**dimensional space) toward which the soil will develop (trajectory) during agropedogenesis and will then remain**
**unchanged within this equilibrium field. Accordingly, the multi-dimensional space of attractors defines the final**
**stage of agropedogenesis.**
The degraded soil will remain within this multi-dimensional space even if subsequently slightly disturbed (or
reclaimed). This explains why long-term agricultural fields that have been abandoned for centuries or even millennia still
show evidence of soil degradation (Hall et al., 2013; Jangid et al., 2011; Kalinina et al., 2013; Lisetskii et al., 2013;
Sandor et al., 2008). For example, abandoned soils under succession of local vegetation such as grassland and forest show
similar physicochemical and biological properties as a result of similarities in their history, i.e. agricultural land-use
(Jangid et al., 2011). The flood-irrigated soils in Cave Creek, Arizona, support only the growth of the Creosote bush even
after about 700 years abandonment. This is in contrast to the presence of seven species of shrubs and cacti in areas
between such soils. The reason is substantial changes in soil texture, i.e. via siltation, thus reducing the water holding
capacity in the flood-irrigated soils and leading to a shift in the vegetation community to more drought-resistant species,
in this case the Creosote bush (Hall et al., 2013). While establishing a no-till system on former pasture-land leads to a
decrease in SOM, changing a formerly plowed land to no-till had no such effect (Francis and Knight, 1993). The amidase
activity in Colca soils, Peru, is still relatively high 400 years after of land abandonment due to the remaining effect of
applied organic amendments on soil microorganisms (Dick et al., 1994). **We argue that during agropedogenesis the**
**multi-dimensional space of master soil properties will continuously narrow in approaching the attractors. This**
**multi-dimensional space resembles a funnel (Fig. 10), meaning that the broad range of all properties in initial**
**natural soils will be narrowed and unified to a (very) small range in agricultural and subsequently degraded soils.**
Identifying the attractors of master properties and the relations among them in this multi-dimensional space yields
diagnostic characteristics to identify and classify agrogenic soils (Gerasimov, 1984).

### 2.7.  Changes in the attractors by specific land-use or climatic conditions

Despite the principle of attractors – the convergence of a property of various soils to one value by degradation – we
assume that these attractors may differ slightly depending on climate, parent material and management. This means that
the multi-dimensional attractor space can have some local minima – metastable states (Kozlovskii, 1999). If the initial
natural soil is close to such a minimum, or the management pushes the trajectory in such a direction, then
agropedogenesis may stop in local minima. Hence, the global minimum will be not reached.
For example, no-till farming may increase SOM in the Ap horizon (Lal, 1997) and level-off at higher values
compared to tillage practices (Fig. 11). However, periodically tilling the soil to simplify weed control quickly destroys the
improvements in soil properties during the no-till period (Cannell and Hawes, 1994). The result is degradation stages
similar to soils under conventional tillage. The ultimate effect of irrigation on soil degradation is expected to be similar to
that of dry-land farming. Despite more organic C input into irrigated systems, the SOM content remains unchanged (Trost



et al., 2014) due to accelerated decomposition (Denef et al., 2008). The state of soil properties in the tropics is predictable
based on pedotransfer functions commonly used in temperate regions, even though tropical soils are usually more clayey;
have lower water holding capacity and a higher bulk density. The explanation lies in the similarities in relations among
soil properties under various climatic conditions (Minasny and Hartemink, 2011). This makes the concept of attractors
generalizable to all cultivated soils (Kozlovskii, 1999), although geo-climatic conditions and specific managements may
modify the attractor values and affect the rates of soil degradation following cultivation (Tiessen et al., 1994).

### 3.  Conclusions and outlook
#### 3.1.  Conclusions
We state that (1) human activities are stronger in intensities and rates than all other soil-forming factors (Liu et al.,
2009; Richter et al., 2015). Because humans exploit mainly one soil function – productivity – they optimize all soil
properties toward a higher yield of a few agricultural crops. And because most crops have similar requirements, the range
of measured values for a given soil property becomes narrower during agropedogenesis. Therefore, human activities lead
to the formation of a special group of agrogenic soils with defined range of properties – Anthrosols. The range of
properties moves toward the attractor specific for each property but the same for different soils. (2) Analyzing the
properties of soils from various geo-climatological conditions and managements in relation to the respective time since
the beginning of cultivation reveals (i) the dynamics of soil properties by agropedogenesis and (ii) demonstrates the final
stage of agrogenic degradation when the values of various soil properties reach the attractor space.
By analyzing the development of soils and the dynamics of soil properties under agricultural use, we develop for the
first time the basic concept of agropedogenesis. This concept is based on (1) the modified classical concept of factors –
processes – properties – functions and back to the processes, (2) the concept of attractors of soil degradation, (3)
identifying master soil properties and analyzing their dynamics by agropedogenesis, (4) analyzing phase diagrams of
master soil properties to identify the thresholds and stages of soil degradation, and finally (5) defining multi-dimensional
attractor space. We defined the attractors and provided the basic prerequisites for elucidating of eight master soil
properties responsible for the trajectory of any soil during agropedogenesis within multi-dimensional attractor space.

#### 3.2.  Outlook
We developed the suggested new concept of agropedogenesis based on the long observation of soil degradation under
agricultural use and on experiments with agricultural soils under various land-use intensities under a very broad range of
climatic conditions. The presented examples of soil degradation trajectories and of attractors of soil properties are clearly
insufficient to reflect the full range of situations. This concept therefore needs to be filled with more observational and
experimental data. Various emerging topics can be highlighted:
Confirmation of master soil properties: The master properties presented here represent suggested entities. This calls
for clarifying whether these are sufficient (or perhaps excessive) to describe the stages of soil degradation under
agropedogenesis. The degree of orthogonality of these properties also remains to be determined. Defining the master soil
properties and their multi-dimensional attractor space will clearly simplify the modelling of degradation trajectories.
Identification of attractor values: The suggested attractor values (Fig. 3, 6, 8b) are mainly based on a few
chronosequence studies and expert knowledge. These values should be defined more precisely based on a broader range



of data. The challenge here is that the average values are probably not optimally suitable as attractors because maximal or
minimal values of a variable are of interest. Therefore, specific statistical methods should be applied, e.g. the border of
the upper (or lower – depending on the property) 95% confidence interval should be used instead of means to set the
attractor value.
The detection of local minima is necessary (and is closely connected with the identification of the multi-dimensional
attractor space). Arriving at such local minima will temporarily stop soil degradation and their determination can be used
to simplify the measures to combat degradation and perhaps even accelerate soil recovery.
Investigating the thresholds and stages of soil degradation, along with identifying the main mechanisms dominating at
each stage, should be done based on the phase diagrams of various soil properties – at least the master properties. These
stages of agropedogenesis with their corresponding main mechanisms are crucial for understanding, modeling, and
combating soil degradation.
Only a few models of natural pedogenesis in its full complexity are available (Finke, 2012; Finke and Hutson, 2008;
Keyvanshokouhi et al., 2016) and the models addressing agropedogenesis describe more or less individual or a selected
few processes of soil degradation. For example various models are available for erosion (Afshar et al., 2018; Arekhi et al.,
2012; Ebrahimzadeh et al., 2018; Millward and Mersey, 1999; Morgan et al., 1998; Pournader et al., 2018; Rose et al.,
1983), SOM decrease (Chertov and Komarov, 1997; Davidson et al., 2012; Del Grosso et al., 2002; Grant, 1997; Liu et
al., 2003; Smith et al., 1997), density increase (Hernanz et al., 2000; Jalabert et al., 2010; Makovnikova et al., 2017; Shiri
et al., 2017; Taalab et al., 2013; Tranter et al., 2007) and other processes due to land-use. Thus, complex theory-based
models of agropedogenesis are required.

**Author contribution**
YK and KZ contributed equally on writing of the paper.

**Competing interest**
The authors declare that they have no conflict of interest.

**Acknowledgements**
This paper is devoted to the 90[th] anniversary of Dr. Sci. Felix I. Kozlovskii – eminent pedologist and geo-ecologist, who
introduced the theory of agropedogenesis more than 30 years ago and was the first to suggest the concept of attractors of
soil degradation.

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





**Table 1: Processes and mechanisms of soil degradation by agricultural land-use**

| | Degradation directions and consequences | Processes and mechanisms | References |
|---|---|---|---|
| Physical properties | Structure:<br>⇩ granular structure<br>⇧ hard clod formation<br>⇧ micro-aggregates and large blocks | - ⇩ SOM content and litter input<br>- aggregate destruction<br>- ⇩ rhizodeposition & mucilage | (Homburg and Sandor, 2011)<br>(Ayoubi et al., 2012; Celik, 2005; Khormali et al., 2009) |
| | Density:<br>⇧ bulk density<br>⇧ subsoil compaction<br>⇧ formation of massive layers | - compaction by heavy machinery<br>- plowing at a constant depth<br>- destruction of aggregates<br>- ⇩ SOM content<br>- ⇩ burrowing animals (earthworms, gophers, etc.)<br>- ⇩ root growth and distribution | (Carducci et al., 2017; Holthusen et al., 2018; Horn and Fleige, 2009; Severiano et al., 2013) |
| | Porosity:<br>⇩ total porosity<br>⇩ water holding capacity<br>⇩ soil aeration | - ⇩ root density<br>- ⇩ burrowing animals<br>- ⇩ large & medium aggregates | (Celik, 2005; Lipiec et al., 2012)<br>(Flynn et al., 2009; Ponge et al., 2013) |
| | ⇩ soil depth | - ⇧ water and wind erosion<br>- ⇧ tillage erosion<br>- ⇧ soil density | (Ayoubi et al., 2012; Govers et al., 1994; Lal, 2001) |
| Chemical properties | ⇩ SOM content<br>⇩ easily available and low molecular weight organic substances | - ⇧ SOM mineralization by increasing aeration<br>- removal of plant biomass via harvesting<br>- residual burning<br>- destruction of macro-aggregates | (Lisetskii et al., 2015; Liu et al., 2009; Sandor and Homburg, 2017) |
| | ⇩ element/nutrient content<br>loss of nutrients<br>narrowing of C:N:P ratio | - removal of plant biomass via harvesting<br>- nutrient leaching<br>- SOM mineralization + NP-fertilization | (Hartemink, 2006; Lisetskii et al., 2015; Sandor and Homburg, 2017) |
| | Acidification:<br>⇩pH<br>⇧exchangeable aluminum<br>⇩CEC | - N-fertilization<br>- cation removal by harvest<br>- ⇩ buffering capacity due to cation leaching and decalcification<br>- acidification and $H^+$ domination on exchange sites<br>- loss of SOM | (Homburg and Sandor, 2011; Obour et al., 2017; Zamanian et al., 2018) |
| | ⇧ salts and/or exchangeable $Na^+$ | - irrigation (with low-quality water or/and groundwater level rise by irrigation) | (Dehaan and Taylor, 2002; Emdad et al., 2004; Jalali and Ranjbar, 2009; Lal, 2015) |



| | | | |
|---|---|---|---|
| Biological properties | | - weeding | |
| | | - pesticide application | |
| | | - monocultures or narrow crop rotations | |
| | ⇓ biodiversity | - mineral fertilization | (Lal, 2009; Zhang et al., 2017) |
| | ⇓ (micro)organism density and abundance | - ⇓ SOM content and litter input | (Breland and Eltun, 1999; Fageria, 2012) |
| | | - ⇓ root amounts and rhizosphere volume | |
| | | - plowing and grubbing | |
| | | - ⇓ total SOM | |
| | | - pesticide application | |
| | | - recalcitrance of remaining SOM | |
| | | - ⇓ microbial abundance activity | (Breland and Eltun, 1999) |
| | ⇓ microbial activities | - ⇓ litter & rhizodeposition input | (Bosch-Serra et al., 2014; Diedhiou et al., 2009; Ponge et al., 2013) |
| | - respiration | - mineral fertilization | |
| | - enzyme activities | - ⇓ organism activity, diversity and abundance | |
| | | - shift in microbial community structure | |
| | | - ⇓ soil animal abundance and activity | |

⇑ and ⇓ means increase or decrease, respectively





**Table 2: Soil formation processes under agricultural practices**

| Additions | Losses | Translocation | Transformation |
|---|---|---|---|
| Irrigation <br> - water <br> - salts ⇧* <br> - sediments | Mineralization ⇧ <br> - organic matter <br> - plant residues <br> - organic fertilizers <br> - nitrogen (to $N_2O$ and $N_2$) ⇧ | Irrigation <br> - dissolved organic matter ⇩ <br> - soluble salts ⇧ | Fertilization <br> - acceleration of nutrient (C, N, P, etc.) cycles <br> - formation of potassium-rich clay minerals |
| Fertilization: <br> - mineral <br> - organic (manure, crop residues) | Erosion: <br> - fine earth erosion ⇧ <br> - whole soil material | Evaporation <br> - soluble salt transportation to the topsoil ⇧ | Mineralization ⇧ <br> - humification of organic residues ⇩ <br> - organo-mineral interactions ⇩ |
| Pest control <br> - pesticides <br> - herbicides | Leaching: <br> - nutrients leaching ⇧ <br> - cations ⇧ <br> - $CaCO_3$ | Plowing/deep plowing <br> - soil horizon mixing <br> - homogenization <br> - bioturbation ⇩ | Heavy machinery <br> - compaction <br> - aggregate destruction ⇧ |
| Amendments <br> - liming <br> - gypsum <br> - sand** <br> - biochar | Harvesting <br> - nutrients <br> - ballast elements | | Pest control <br> - fungal community ⇩ |

* ⇧ and ⇩ imply the increase or decrease, respectively, in rates of processes that may also occur under natural conditions
** To improve soil texture and permeability





**Table 3: Soil properties suggested in the literature as being master properties**

| Suggested minimum set of master properties | References |
|---|---|
| Clay content, CEC, bulk density | (Minasny and Hartemink, 2011) |
| CEC, CaCO$_3$ content, Exchangeable sodium percentage (ESP), Sodium absorption ratio, pH | (Nabiollahi et al., 2017) |
| Bulk density, Mg content, Total N, C:N ratio, Aggregate size distribution, Penetration, Microbial respiration | (Askari and Holden, 2015) |
| Labile phosphorus, Base saturation, Extractable Ca | (Lincoln et al., 2014) |
| C:N ratio, Labile phosphorus, C$_{humic}$:C$_{fulvic}$, Gibs energy, SiO$_2$:(10R$_2$O$_3$) | (Lisetskii et al., 2013) |
| pH, Sodium absorption ratio, Potentially mineralizable N, Labile phosphorus | (Andrews et al., 2003) |
| Labile (active) carbon | (Bünemann et al., 2018) |
| Microbial biomass, Microbial respiration | (Guillaume et al., 2016b) |
| pH, Arylsuphatase activity | (Raiesi, 2017) |
| Geometric means of microbial and enzyme activity | (Raiesi and Kabiri, 2016) |
| Coarse fragments, pH, SOC, total N, ESP, exchangeable cations (Ca, Mg, and K), and available phosphorus | (Rezapour and Samadi, 2012) |
| Physical: Bulk density (1.7 g cm$^{-1}$), Macroaggregates (0%), Soil depth (A+B horizons 20 cm) | |
| Chemical: SOM content (50% of natural), C/N (8-10), pH (4 or 10), EC (16 dS m$^{-1}$)* | This study** |
| Biological: Microbial biomass C, Basal respiration | |

* CEC has been omitted from chemical master properties because it depends on (i) clay content and clay mineralogy – whose
properties are resistant to agricultural practices, and (ii) SOM, which is considered a master property.
** The values in brackets are very preliminary attractors of each property by anthropogenic soil degradation. The two pH attractors are
presented for acidic (humid climate) and alkaline (semiarid climate) soils. Note that not all attractors can be suggested in this study.
The criteria for selecting master soil properties are described in the text.





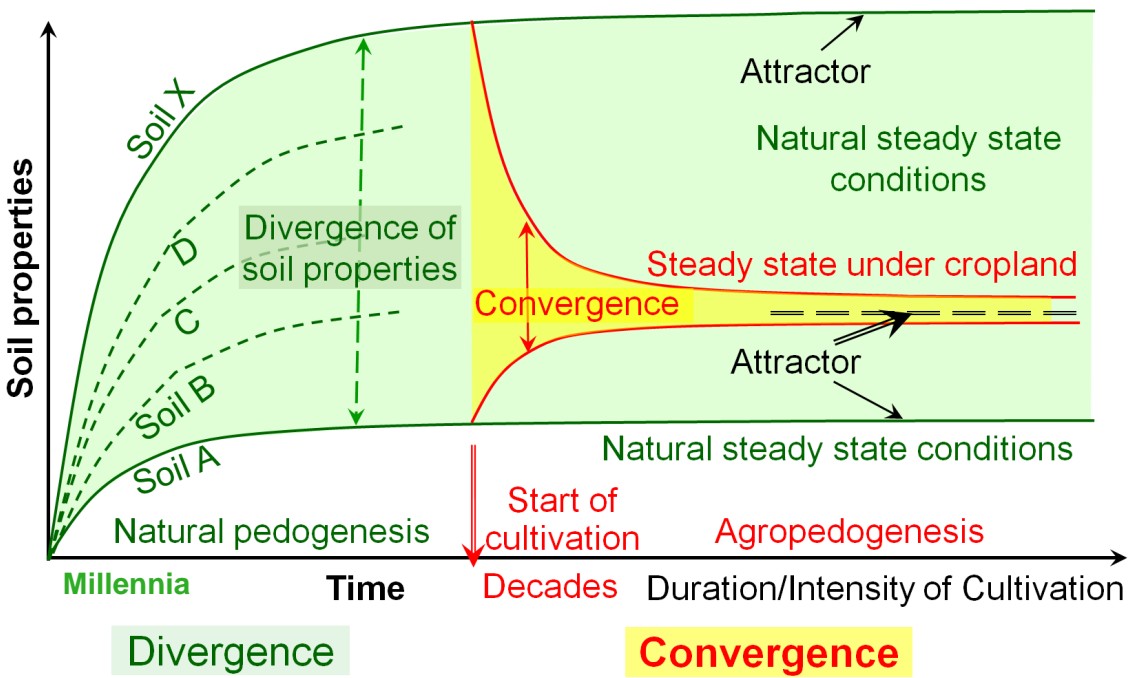

Fig. 1: Conceptual scheme of soil development, i.e. pedogenesis, under natural conditions (green lines) and agropedogenesis due to long-term agricultural practices (red lines). Natural pedogenesis leads from the initial parent material to a wide range of steady state values (green arrow) for a given soil property over hundreds or thousands of years due to various combinations of the five soil-forming factors. Natural pedogenesis leads to *divergence* of soil properties. In contrast, agricultural practices and the dominance of humans as the main soil-forming factor cause each property to tend toward a very narrow field of values, i.e. attractors of that property defined by human actions, namely land management for optimization of crop production. Therefore, agropedogenesis leads to *convergence* of soil properties.





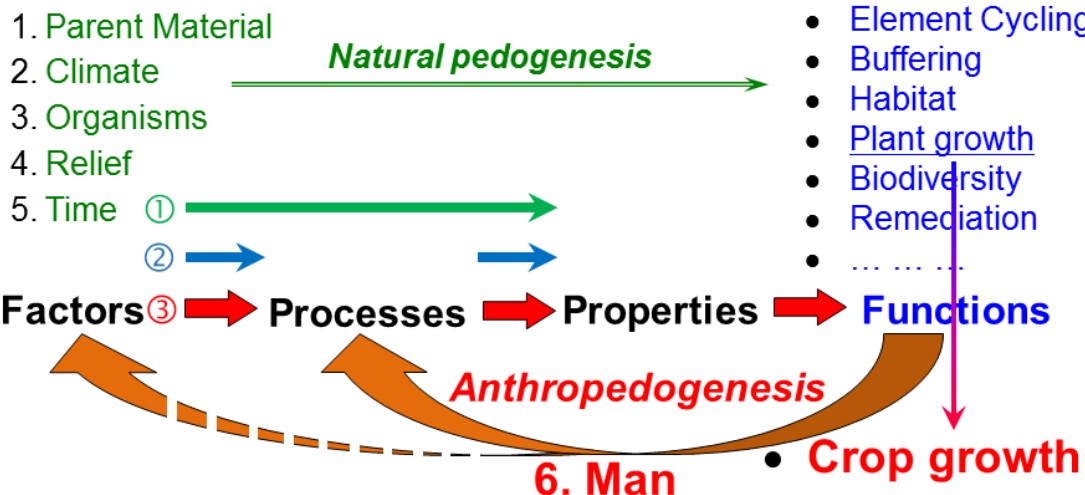

Fig. 2: Soil genesis based on the development of concepts: ① 'Factors → Properties' (Dokuchaev, 1883; Jenny, 1941) –
green arrow, ② 'Factors → Processes → Properties' (along the blue arrows) (Gerasimov, 1984), ③ our introduced
concept 'Factors → Processes → Properties → Functions' (along the red arrows). The latter concept considers not only
the functions of natural soils, but especially human modification of soils toward only one function of interest (here, crop
growth). Anthropogenic optimization of only one function involves strongly modifying processes and factors, leading to
formation of a new process group: Anthropedogenesis.




Fig. 3: Examples for attractors of soil properties by anthropogenic degradation: (a) Soil organic carbon content, (b) Total
nitrogen content, (c) Infiltration rates, (d) Exchangeable Ca and Mg content, (e) C to N ratio in soil particles, and (f)
overall decrease in soil quality, i.e. degradation over cultivation period. Yellow shading: area covered by all experimental
points, showing decrease of the area with cultivation duration. Blue double arrows: range of data points in natural soils
(left of each Subfig.) and strong decrease of data range due to cultivation.
(a) Narrowing range (blue arrows) of soil organic carbon over cultivation periods in southern Queensland, Australia (6
sites) (Dalal and Mayer, 1986a) and savanna soils in South Africa (3 sites) (Lobe et al., 2001). The natural soils in





different climatic regions have various ranges of properties, e.g. organic carbon from 0.8-2.3%. During cultivation however, the organic carbon content strongly narrows to between 0.3-1.0%.

(b) Narrowing range (blue arrows) of total soil nitrogen over cultivation periods. Sampling sites similar as (a) plus 5 sites (hexagon symbols) from Great Plains, USA (Haas et al., 1957). Before commencing agriculture, the Great Plains soils had a wide range of texture classes (silt loam, loam, clay loam, and very fine sandy loam), an initial organic carbon content of 1.13-2.47%, and a total nitrogen content of 0.05-0.22%. Nonetheless, the total nitrogen range narrowed to 0.03-0.07% over 45 years of intensive agriculture. As (Haas et al., 1957) anticipated, all soils may finally reach a similar value for total nitrogen (i.e. the attractor of nitrogen) by continuing the ongoing management (in line with Australian and South African soils).

(c) Infiltration rates as a function of years since land-use change from forest to agriculture (Nyberg et al., 2012). Note the narrowing trend (the blue arrows) in measured values from forest (t = 0) toward long-term cultivations (t = 39, 57, 69 and 119 years since conversion). The measured value at ca. 120 years is defined as the attractor of the infiltration rate, and 120 years is the time needed to reach that attractor.

(d) Narrowing content (blue arrows) of exchangeable Ca and Mg in the first 15 cm of Oxisols during 31 years (1978-2009) of sugar cane cultivation (Morrison and Gawander, 2016). The three soils developed under different natural vegetation prior to cultivation and received different managements thereafter.

(e) Narrow ranges of C:N ratios in all texture classes (sand, silt and clay) over 85 years of cultivation (Eleftheriadis et al., 2018). Note the different rates of C:N decrease in the three fractions. That ratio in the sand fraction is more susceptible to cultivation duration, but is rather resistant in the clay fraction.

(f) Dependence of soil quality index on duration and intensity of soil cultivation (on the x-axis: 1- Virgin land, 2- Idle land in the modern era, 3- Modern-day plowed land, 4- Post-antique idle land, 5- Continually plowed land) over 220 to 800 years cultivation (Lisetskii et al., 2015). Note that soil quality became similar (blue arrows) with increasing cultivation duration and/or cultivation intensity (from 1 to 5) (Value in red circle is an outlier).





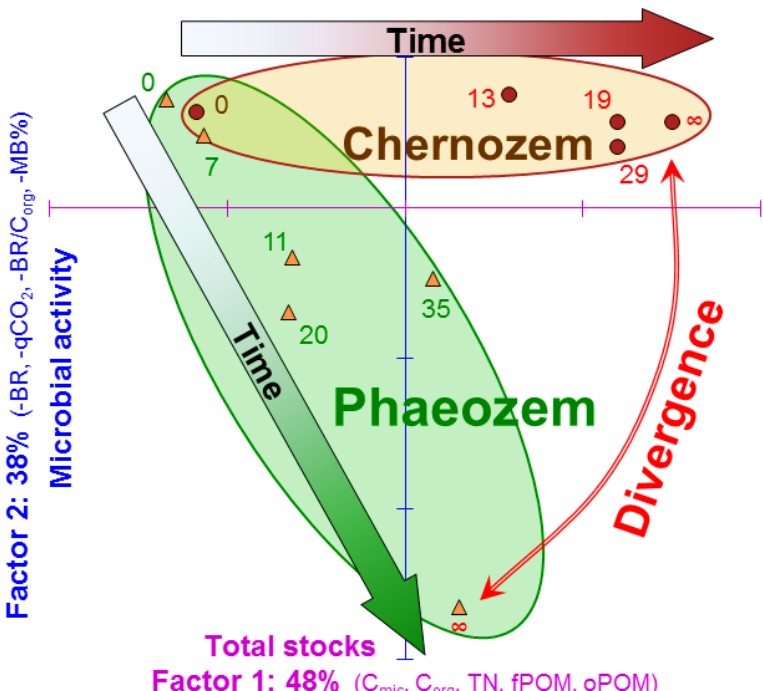


Fig. 4: Divergence of properties of agriculturally used Chernozem (CH) and Phaeozem (PH) after abandonment analyzed
by principal component analysis (PCA, Kurganova et al., 2019, submitted). The soils had very similar properties due to
long-term (> 100 years) cropping. After abandonment, they started to develop to their natural analogues ($\infty$), leading to
strong divergences of their properties. This figure reflects the divergence, i.e. the opposite situation to agricultural use.
Numbers close to points: duration of abandonment, 0 is agricultural soil and $\infty$ is natural analogues (not cultivated). The
soil parameters primarily driving the divergence are: microbial biomass C ($C_{mic}$), soil organic C ($C_{org}$), total N (TN), free
particulate organic matter (fPOM), occluded organic matter (oPOM), basal respiration (BR), metabolic coefficient
($q$CO$_2$), BR/$C_{org}$ ratio, and portion of microbial biomass (MB%).



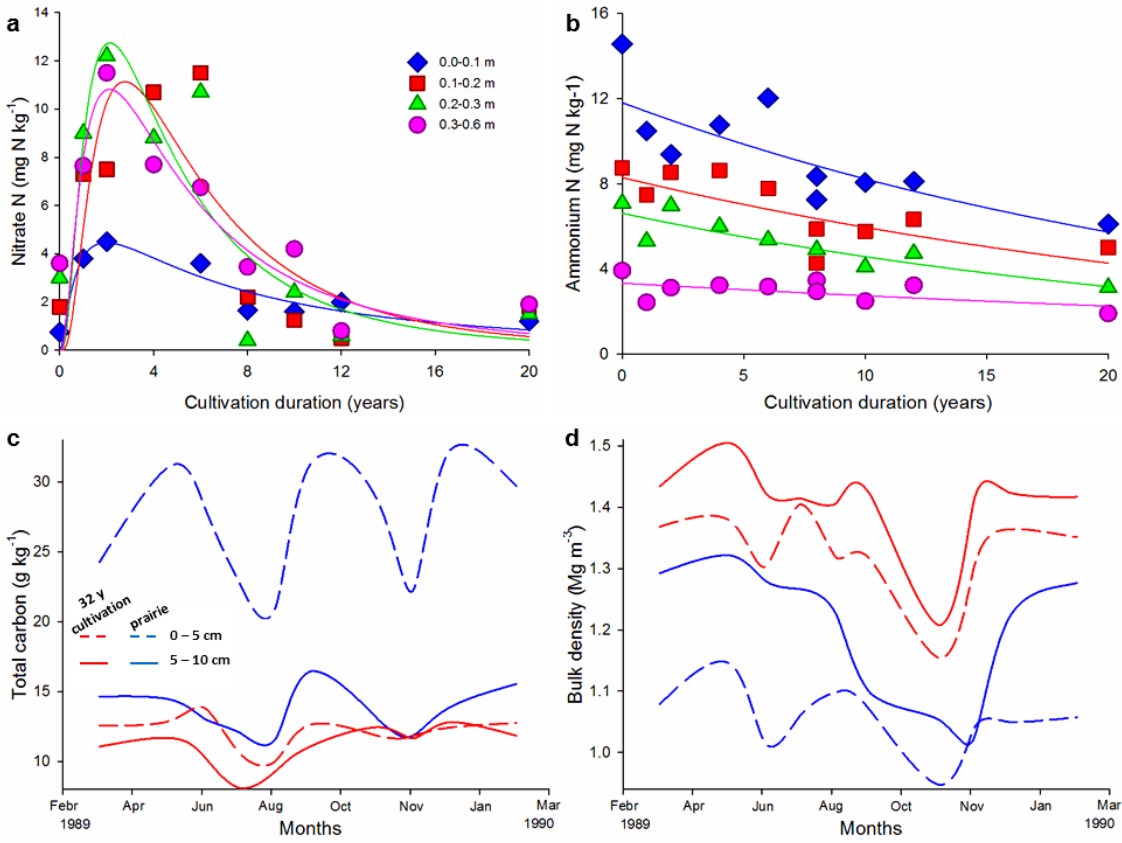


Fig. 5: Homogenizing effects of cultivation and cultivation duration on soil properties. (a) Nitrate N, (b) ammonium N
contents depending on soil depth during 20 years of cultivation (Jones and Dalal, 2017), (c) and (d) total soil carbon and
bulk density, respectively, during one year in two soil depths under natural prairie and after 32 years of cultivation (Scott
et al., 1994).





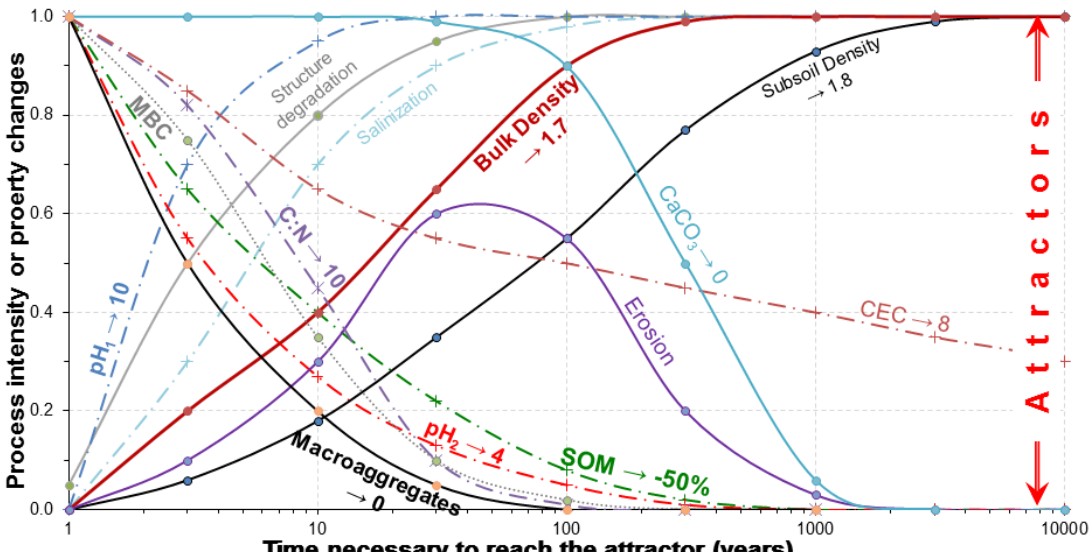


Fig. 6: Overview on rates of key processes of agropedogenesis and their trajectory in reaching their attractors. Curves
start from 0 or 1 at the onset of cultivation and go to 1 or 0 to the specific attractors. Each curve labeled with the specific
property. Small arrows: estimated level of attractor. Curve shape, time to reach attractor, and attractor levels are estimates
and require future adjustment based on experimental data. $pH_1$ is for alkaline, $pH_2$ for acidic soils. Note that not all
attractors are defined yet. Properties in bold: master soil properties for agropedogenesis. MBC: microbial biomass carbon,
SOM: soil organic matter, CEC: cation exchange capacity. Continuous lines present physical properties or processes, dot-
dashed lines correspond to chemical, dotted lines to biological properties.





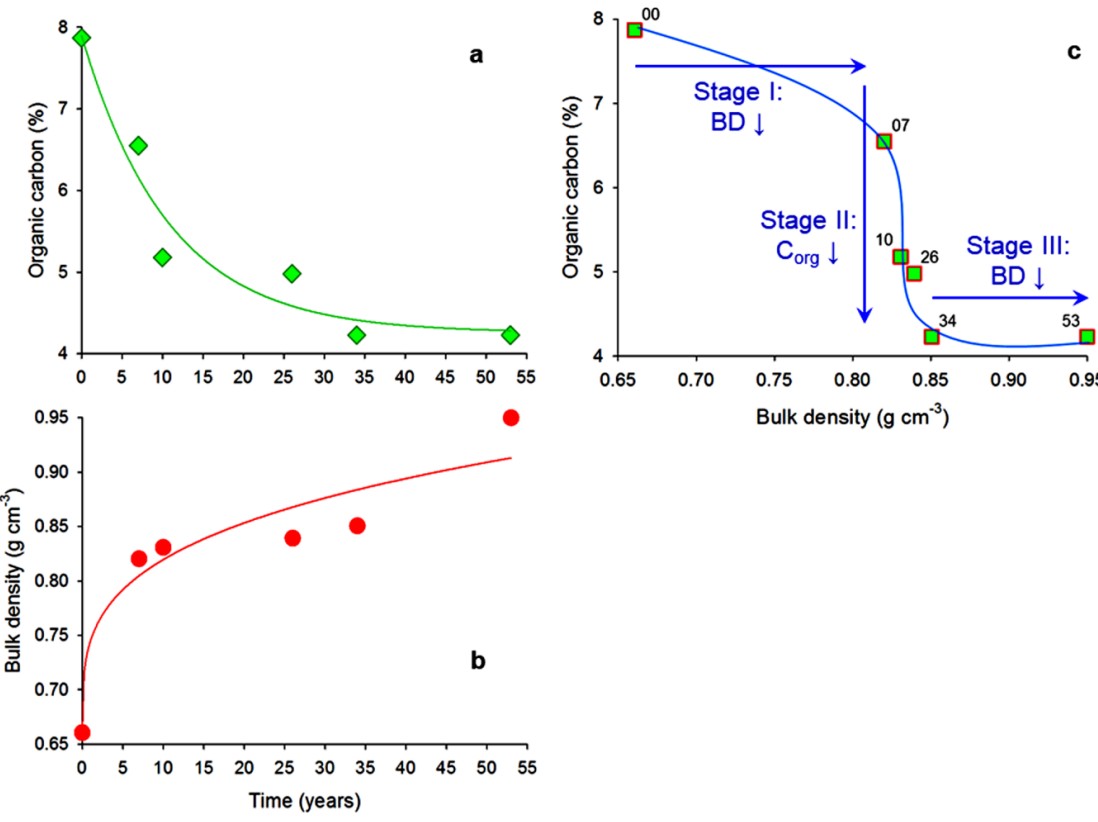


Fig. 7: Effects of duration of forest conversion to cropland on decreasing soil organic carbon (a) and increasing bulk
density (b) during 53 years (Southern Highlands of Ethiopia, (Lemenih et al., 2005). (c) Phase diagram: relation between
SOC and bulk density at corresponding time. Note the stepwise changes in bulk density following decreasing SOC
content below the thresholds of 7.8, 6.5 and 4.2%. Numbers beside symbols refer to years after conversion.





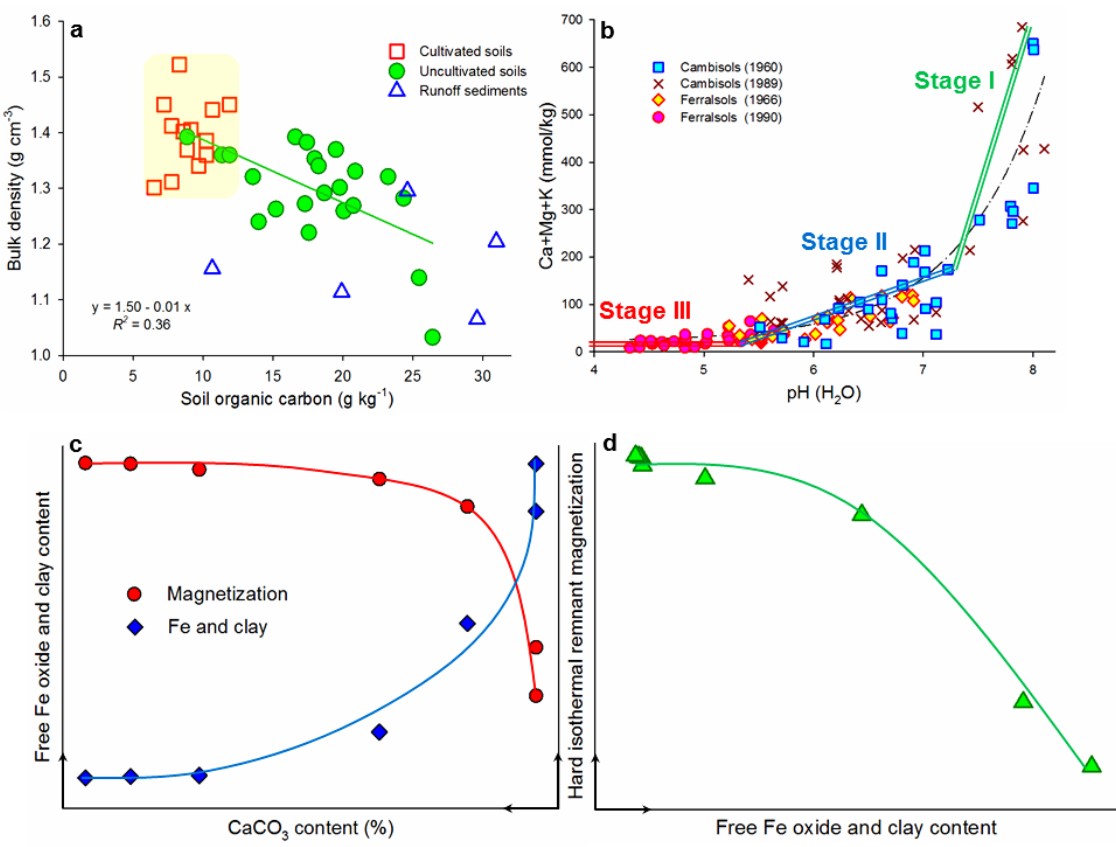


Fig. 8: Phase diagrams of various properties of agricultural soils. Small arrows at the start or end of the axes show the
increase of corresponding soil property.
(a) Narrow range (yellow-shaded area) of organic carbon and bulk density in ancient agricultural soils cultivated for 1500
y at Mimbres, New Mexico, USA, comparing to uncultivated soils and runoff sediments (Sandor et al., 2008). Note that
the decreasing trend of bulk density with increasing soil organic carbon content (green line with regression equation for
uncultivated soils) is absent in cultivated soils (Sandor et al., 2008).
(b) Changes in exchangeable base cations depending on soil pH in Cambisols and Ferralsols in coastal plains of Tanzania
(Hartemink and Bridges, 1995). Ferralsols clearly decline in exchangeable cations (i.e. two separated groups in phase II
and III) with decreasing pH over ca. 24 years of cultivation. The exchangeable cations in Cambisols remain in stage I.
Double lines: stages of exchangeable cation decrease with decreasing soil pH. Content of exchangeable cations levels off
at ~ 25 mmol+ kg-1 (stage III). This value – which corresponds to the amount of exchangeable $Ca^{2+}$ and $Mg^{2+}$ shown on
Fig. 3d (31 years of sugar cane cultivation on Fijian Ferralsols) – is an attractor.
(c) The content of free iron oxides, clay content and hard isothermal remnant magnetization (IRMh) as a function of
$CaCO_3$ content in soil (adopted from (Chen et al., 2011).).
(d) The relation between IRMh and free iron oxides vs. clay content.





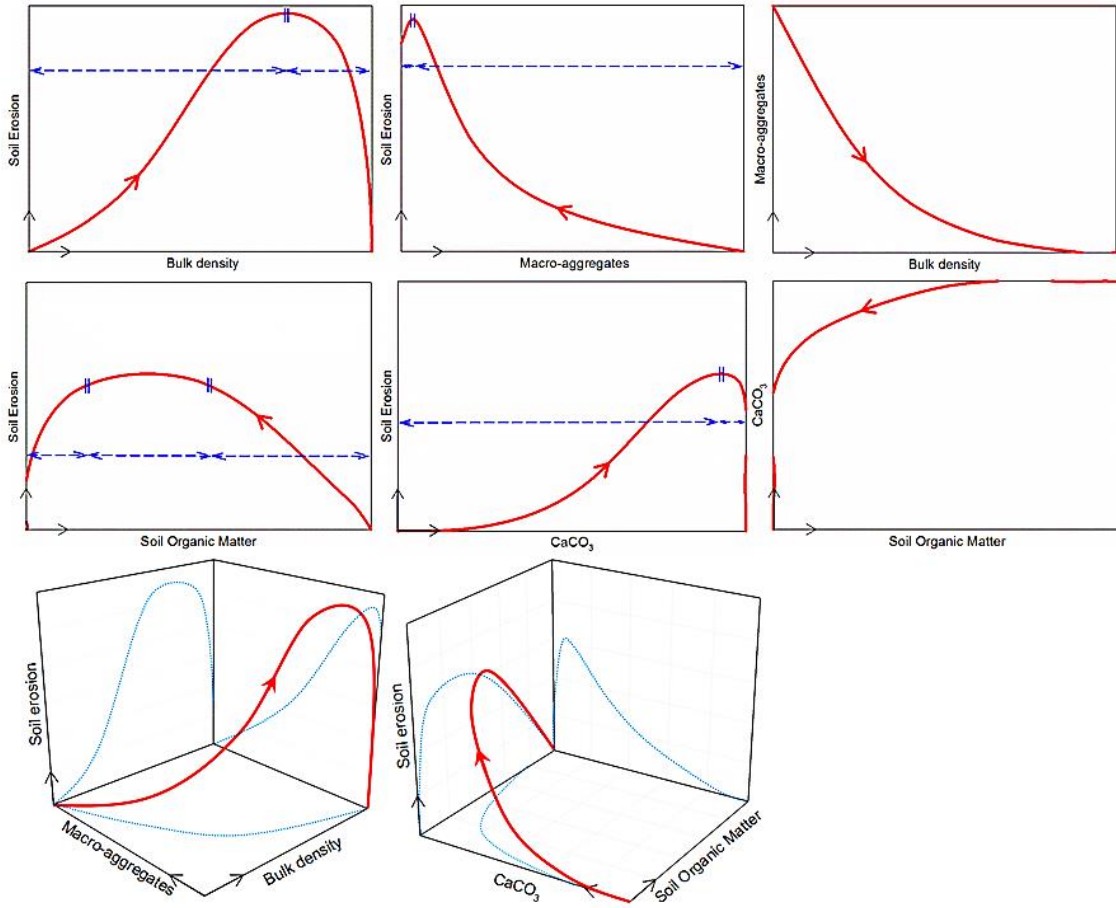

Fig. 9: Examples of conceptual 2D and 3D phase diagrams linking soil erosion intensity with (top) bulk density and macroaggregates content, (middle) SOM and CaCO$_3$ contents during agropedogenesis. Small red arrows on curved lines: direction of soil degradation. Horizontal blue dashed arrows show the stages, and vertical blue double lines show the arbitrary thresholds of soil degradation. Projections of 3D lines (light blue) on last Subfigures (bottom) correspond to the individual lines on the 2D phase diagrams in top and middle. Similar phase diagrams can be built in multi-dimensional space corresponding to the number of the master soil properties.





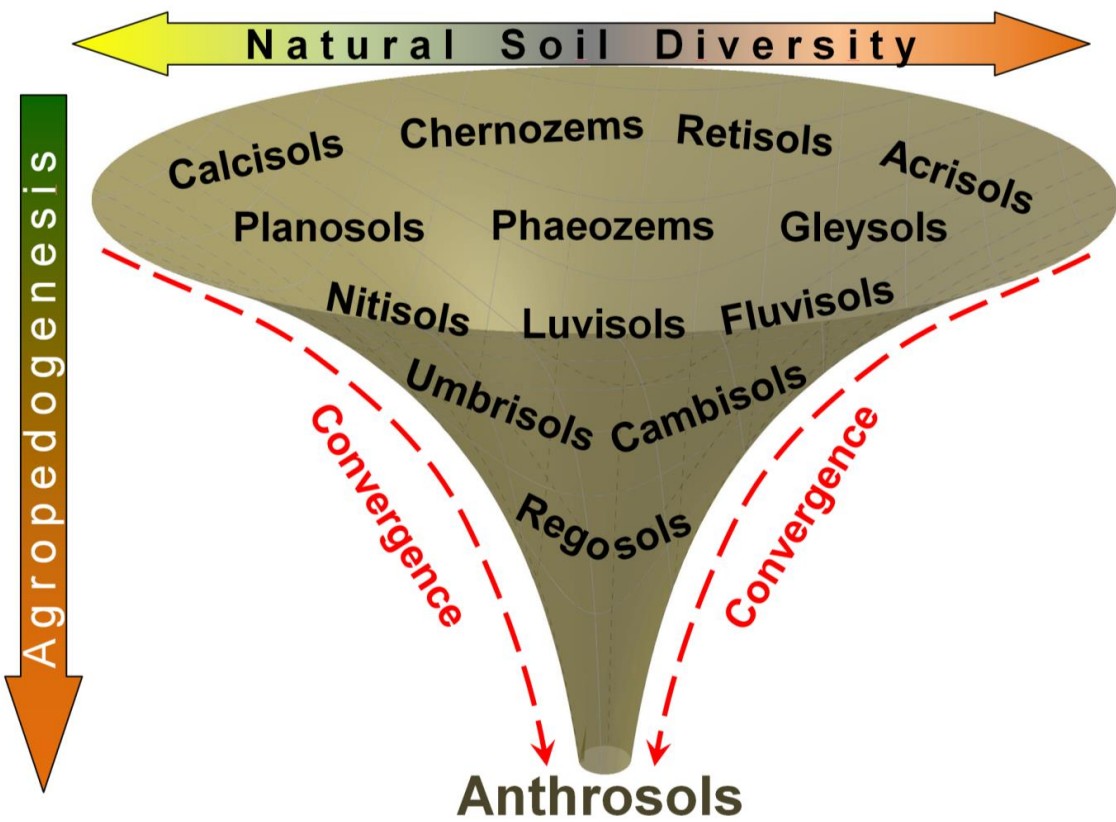

Fig. 10: Conceptual schema of convergence of soil properties by agropedogenesis. The very broad range in natural soils

will be tailored for crop production by agricultural use, resulting in Anthrosols with a very narrow range of properties.



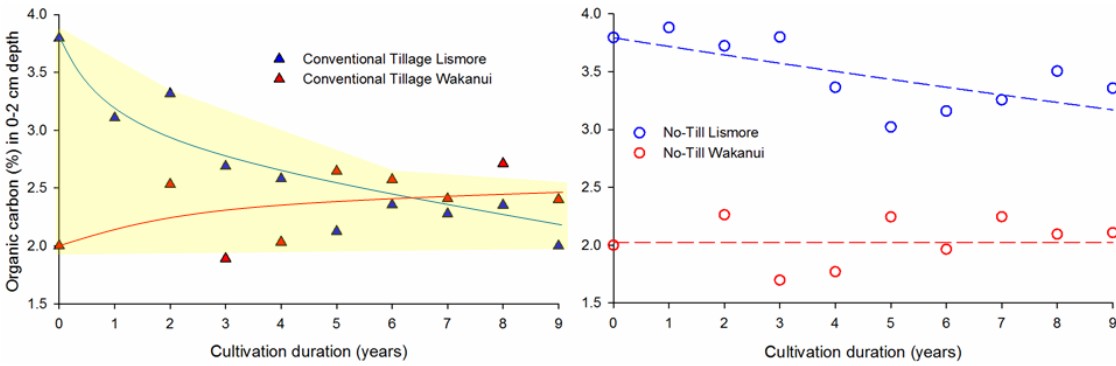

Fig. 11: Nine years of continuous cropping and conventional tillage (left) led to similarities in soil organic carbon content in contrast to no-till soils (right) (Francis and Knight, 1993). The Lismore no-till soil either needs longer cultivation duration to reach the carbon content characterizing soils under conventional tillage or the attractor of SOC has already reached, i.e. local minima for this soil. Note that the Wakanui no-till soil was cultivated for 10 years before beginning the trial and thus show similar values, i.e. similar attractor for SOC as under conventional tillage. Hence, changing the conventional tillage to no-till had no effect on organic carbon content. Lismore soil: Umbric Dystochrept, 5% stones, rapid draining, 5 y mixed rye grass/white clover pasture. Wakanui soil: Udic Ustochrept, slow draining, 10 y rotation of wheat, barley, peas.