# Peer review of "Title: Agropedogenesis: Humankind as the 6th soil-forming factor and attractors of"

_Biogeosciences, 2019_

## Short Comment (SC1) · 15 Jun 2019

CommentsâČř1,2 on the publication entitled "Agropedogenesis: Humankind as the 6th soil-forming factor and attractors of agrogenic soil degradation" by Yakov Kuzyakov and Kazem Zamanian in ''Biogeosciences Discuss., https://doi.org/10.5194/bg-2019-151. Manuscript under review for journal Biogeosciences. Discussion started: 8 May 2019. âČřBy 1.Dr. D. K. Pal, Former Principal Scientist, Division of Soil Resource Studies, ICAR-NBSS&LUP, Amravati Road, Nagpur 440033, India. Email: paldilip2001@yahoo.com 2. Dr. Ashim Datta, Scientist, Division of Soil and Crop Management, ICAR-CSSRI, Karnal 132001, India. Email: ashimdatta2007@gmail.com

Comments are made on three important aspects based on research made on Indian tropical soils. (a)The first one is on the accumulation of soil organic carbon (SOC) by growing agricultural crops in soils of semi-arid tropical (SAT) climate of India showing no sign of soil degradation. (b)The second one is on the resilience of SAT sodic soils through anthropogenic activities (6th factor of soil formation) showing pedogenic processes that are reverse to what was proposed in the conceptual model of agropedogenesis. (c) The third one is to compete with the idea of acidification (lowering of soil pH) under agricultural crops is considered as a sign of soil degradation in model concept of agropedogenesis. (a) Soils as a unique natural capital, provide multiple benefits to humans, and are the foundation of agriculture and deliver multiple soil ecosystem services. It is expected that global food production will grow by 50% by the year 2030 and by 100% by the year 2050 in order to fulfil the demands of a growing world population and changing food consumption patterns (Godfray et al., 2010). It is often feared that the current long-term agricultural systems may damage the natural capital. Therefore, a need is always felt to mitigate the negative effects of agriculture on soils through 'sustainable intensification of agriculture', which would maintain the increasing trend of the current agro-ecosystems by sustaining natural capital stocks and also by minimizing adverse effect on environment (Baulcombe et al., 2009; Godfray et al., 2010; Koch et al., 2013; Lal, 2009). Projections indicate that 80% of crop production growth in developing countries up to 2030 will be derived through intensification (FAO, 2009). However, there are evidences to indicate that increasing agricultural intensification can erode ecosystem services (Power, 2010; Tilman et al., 2001, 2002). Continuous agriculture as the powerful anthropogenic activities by human beings to produce more food stocks, may thus cause degradation of soils, which Kuzyakov and Zamanian (2019) described as ''Agropedogenesis'', and these authors consider this degradation as the 6th soil -forming factor. They have identified a set of 'master properties' (bulk density and macroaggregates, soil organic matter content and pH, microbial biomass and basal respiration), which are especially sensitive to land use and determine the other properties during agropedogenesis. It is an emerging conceptual model in soil science and

thus stands for its universal acceptability by highlighting some case studies where several decades and century long agricultural practices are prevalent in the world history of agriculture especially when the intensification is being realized without degrading soil natural capital and ecosystem services. Halting of degradation of soils demanded the practices to change to more sustainable practices (Baulcombe et al., 2009) by using the non-renewable resources to renewable inputs (Sandhu et al., 2015), such as organic fertiliser (manure, compost), renewable energy and biologically based technologies for pest control. Such shift in practice is already under way in tropical soils of many agriculturally based countries including Indian sub-continent. Therefore, it will be important to showcase and also to share some soil carbon research accomplished in India following the recommendations of the National Agricultural Research System (NARS), which suggests no sign of soil degradation in terms of agropedogenesis as conceptualized and proposed by Kuzyakov and Zamanian (2019). It is realized that the carbon content in soils changes depending on the land use system and time. There is an increasing concern all over the world about the decline in soil productivity and the impoverishment of soil organic carbon (SOC) caused by intensive agriculture. The National Bureau of Soil Survey and Land Use Planning (NBSS&LUP) of the Indian Council of Agricultural Research (ICAR), through organized research initiatives, sponsored by national and international organizations, has developed datasets of SOC for two important crop production zones, viz. the Indo-Gangetic Plains (IGP) (52.01 M ha, occupied mainly by Entisols, Inceptisols, Alfisols, Mollisols, Vertisols and their intergrades, Pal, 2017) and the black soil region (BSR) (76.4 M ha, occupied mainly by Vertisols and their intergrades and Alfisols, Pal, 2017) in the semi-arid tropics (SAT) (Bhattacharyya et al., 2014; Mandal et al., 2014). The datasets for 1980 and 2005 indicate an overall increase in SOC stock in the Benchmark spots under agriculture, practised for the last 25 years. This suggests that the agricultural management practices advocated through NARS for the last 25 years did not cause any decline in SOC in the major crop growing zones of the country (Bhattacharyya et al., 2007). Results of long-term fertilizer experiments with rice-based double or triple cropping systems indicate soil's capacity

to store greater C, and maintain higher C in passive pools and that active fraction of soil C can be used as an indicator of soil health. The application of NPK plus FYM emerged as a cost-effective technology for Indian farmers (Pathak et al., 2011; Mandal et al., 2008; Datta et al., 2017, Krishna et al., 2018) because the application of organic amendments builds up the SOC pool which enhances the quality of the soil, ecosystem services of soils, water and combats climate change (Lal, 2011; Christensen et al., 2009). However, management interventions of the NARS have caused depletion of soil organic carbon in some Mollisol of the IGP and they are presently classified as Alfisols having no other form of degradation. These Alfisols' ecosystem service in producing bumper wheat crop is still maintained. Additionally, to produce bumper rice, wheat and potato crops the IGP soils under cultivation for the last three decades used all modern agricultural implements and irrigation, which caused a rise in bulk density (BD) in the subsoils (Chandran et al., 2014; Tiwary et al., 2014). Although the rise in BD is responsible for plateauing or decline in the yield of the subsequent crop like wheat, it helped in maintaining yield of rice in rainy season and also in sequestering more SOC under sub-merged condition (Bhattacharyya et al., 2007; Sahrawat et al., 2005; Mandal et al., 2008; Pal et al., 2015). Paradoxically, the short and long-term experiments on Indian tropical soils reported during the past decades and in recent times indicate that these soils have reached a quasi-equilibrium stage and seldom show OC content (0-30 cm) > 1%. It is noted that even the SOC content (0-30 cm depth) of a long-term experiment (LTE) of 28 years on SAT Vertisols (with clay smectite $\sim$ 90%) using sorghum-wheat cropping system with recommended doses of NPK fertilizers plus FYM (10t ha-1) show a value of 0.73% only (Datta et al., 2018). The increase in OC in SAT agricultural soils (maximum up to $\leq$ 1% in 0-30 cm depth) through NARS interventions (Bhattacharyya et al., 2007, 2008) appears enough to provide ecosystem services in growing self-sufficiency in food production and food stocks since independence. The NARS interventions since post-green revolution period have helped in increased OC sequestration in all soil types and have not caused any degradation as evidenced from the retention of their original US Taxonomic soil orders (Pal, 2017). Also, such soils did

not lead to increased emissions of greenhouse gases to any alarming proportion (Pal et al., 2015). However, it is intriguing that why the SAT soils (non-acidic, calcareous and dominated either by mixed or smectitic clay minerals) under decades long NARS management for agricultural crops, do not have OC > 1%? This scenario affirms that anthropogenic interventions by humankind in tropical soils of India under SAT environment are far from being qualified to be the 6th soil factor of soil formation in the model of 'agropedogenesis', proposed by Kuzyakov and Zamanian (2019). (b) Canal irrigation was introduced at the end of the 19th century to minimize the problem of aridity and to stabilize crop yields in the north western part of the IGP. This resulted in the expansion of the cultivated area. However, introduction of irrigation during the dry climate without the provision of drainage led to soil salinization and alkalinisation within a few years, due to rise in the groundwater table containing high proportion of sodium relative to divalent cations and/or high residual alkalinity. In addition, the use of groundwater with high sodicity hazards for irrigation has resulted in the extension of sodic soils (Abrol, 1982a, b). Apart from these kinds of salt-affected soils as an effect of anthropogenic activities, sodic soils interspersed with non-sodic or less sodic soils also occur in unirrigated areas of the SAT parts of the IGP and BSR regions. Therefore, anthropogenic activities in the IGP and BSR areas are not the only reason for the development of sodic soils (Pal et al., 2009a, b, 2016). The soils of the SAT regions in general are calcareous and, on many occasions, they are also sodic either in the subsoil or throughout the depth of soil profile (Pal et al., 2000; 2006). Calcareousness of these soils is due to the pedogenic formation of calcium carbonate. The formation of pedogenic calcium carbonate (PC) in the arid climate enhances the pH and also the relative abundance of $Na^+$ ions on soil exchange sites and in the solution; and the $Na^+$ ions in turn cause dispersion of the fine clay particles. The dispersed fine clays translocate in soils as the formation of PC creates a $Na^+$-enriched chemical environment conducive for the deflocculation of clay particles and their subsequent movement downward. Therefore, the formation of PC and the clay illuviation are two concurrent and contemporary pedogenetic events, resulting in an increase in relative proportion of sodium, causing

increased sodium adsorption ratio (SAR) and exchangeable sodium percentage (ESP) and pH values with depth. Thus, the formation of PC is a basic natural degradation process, which exhibits the regressive pedogenesis by capturing atmospheric $CO_2$ (Pal et al., 2013; 2016). Sodic black soils (Sodic Haplusterts), which are impoverished in OC but are rich in $CaCO_3$, show enough resilience under improved management (IM) system (without adding gypsum and FYM) of the International Crops Research Institute for Semi-Arid Tropics (ICRISAT). The average grain yield of the IM system over thirty years was five times more than that in the traditional management (TM) system (Wani et al., 2003). Due to the improvements in physical, chemical and biological properties of soils after adaptation of the IM system, the poorly drained black soil (Sodic Haplusterts) now qualify for well drained soils (Typic Haplusterts). Continuous release of higher amount of $Ca^{2+}$ ions during the dissolution of $CaCO_3$ under the IM system, compared to slower rate of formation of $CaCO_3$, provide enough soluble $Ca^{2+}$ ions to replace unfavourable $Na^+$ ions on the soil exchange sites. Higher exchangeable Ca/Mg ratio in soils under IM system improved the saturated hydraulic conductivity (sHC) for better storage and release of soil water during the dry spell between rains. Adequate supply of soil water helped in better crop productivity and higher OC sequestration (Pal et al., 2012a, b). The improvement in Vertisols' sustainability suggests that the IM system is capable of mitigating the adverse effect of climate change. This management protocol though slow as compared to the gypsum-aided one, is however cost-effective and farmer-friendly. This technology helps to realize the benefit of the presence of $CaCO_3$ as an ecosystem engineer during the reclamation of sodic soils. Role of PC as an ecosystem engineer is also evidenced during the reclamation processes of IGP sodic soils even after the addition of gypsum. Sodic soils (Natrustalfs) of NW part of the IGP, after their reclamation by gypsum, improve in terms of their morphological, physical and chemical properties so much that these soils are now reclassified as well-drained and OC-rich normal Alfisols (Haplustalfs). Significant improvement in SOC stock and other soil properties was observed under different land uses in reclaimed alkali soils of North West India (Datta et al., 2015). Such remarkable resilience could

be possible even with the low amount of added gypsum, suggesting that the added gypsum does not enrich soil solution by the required amount of Ca2+ ions to replace Na ions on the soil exchange sites. The fulfilment of Ca saturation could be possible by the dissolution of PC during the growing of the rice crop under submerged conditions (Pal et al., 2016). It was observed that the rate of dissolution of PC was much higher than its rate of formation in the top 1.0 m soil depth (Pal et al., 2009a; 2012b; 2016). Even after becoming normal soils as Haplustalfs and Haplusterts both IGP and Vertisols are still calcareous. In view of its slow rate of dissolution, it is quite likely that Ca-ions enriched chemical environment would allow neither Haplustalfs nor Haplusterts to transform to any other soil order so long CaCO3 would continue to act as a soil modifier. Positive role of CaCO3 in both the reclamation and sequestration of OC in SAT soils may benefit the maintenance of soil health of the farmlands (Pal et al., 2016). The above discussion makes it clear that anthropogenic interventions to make sodic soils resilient and also sustainable for production of agricultural crops is a unique case of 6th factor of soil formation which contrasts the degradation process of the agropedogenesis model proposed by Kuzyakov and Zamanian (2019). (c) For a long time, acid soils were considered to be chemically degraded because of their high acidity (pH «6.5), which is caused by the profuse chemical weathering under humid tropical (HT) climate. Moreover, they are often conceived to be typical soils that have less soil fertility generally, which however strongly contrasts with their OC enrichment (> 1%) in the surface horizons (Pal et al., 2014). They occupy about 9.4% of the total geographical area of the Indian sub-continent (ICAR-NAAS, 2010). Soils of HT climate in the states of Kerala, Goa, Karnataka, Tamil Nadu and North East Hill areas are strongly to moderately acidic Alfisols, Ultisols and Mollisols and their further weathering in HT climate would finally close at kaolin dominated soils with considerable amount of layer silicate minerals (Pal et al., 2014) and thus are siliceous in nature (especially the Ultisols of Kerala and acidic Alfisols of Goa) (Chandran et al., 2004; 2005; Varghese and Byju, 1993). This suggests that silica is insoluble in acidic soil medium and thus causes an incomplete desilication process in these acidic HT soils. It is interesting to note that the

amount of SiO2 and its molar ratios of Ultisols of Kerala (Chandran et al., 2005) and acidic Alfisols of Goa (Chandran et al., 2004) are comparable with some of the Oxisols reported from Puerto Rico (Jones et al., 1982), Brazil (Buurman et al., 1996; Muggler, 1998), and other regions of the World (Mohr et al., 1972). However, in the acidic Alfisols, Ultisols and Mollisols, the process of desilication no longer operates in present day conditions because the pH of the soils is well below the threshold of 9.0 (Millot, 1970). Interestingly, in such million years old soils, desilication and transformation of kaolin to gibbsite is pedogenetically impossible (Chandran et al., 2005; Pal, 2017) and thus it will be an equally impossible proposition for further degradation of Ultisols: they would however continue to provide ecosystem services in supporting agriculture, horticulture, forestry, tea, coffee and spices (Sehgal, 1998; Pal et al., 2014, Pal, 2017). This novel insight however contrasts with the representation of degradation of Ultisols to agriculturally unproductive Oxisols at the last stage of advanced stage of soil weathering (Smeck et al., 1983; Lin, 2011) and finally reaching the thermodynamic equilibrium. The formation and persistence of HT soils on the other hand, provide an example that in an open system such as the soil, the existence of a steady state seems a more useful concept than based on equilibrium in a rigorous thermodynamic sense (Bhattacharyya et al., 1999; 2006, Chandran et al., 2005; Pal, 2017). The present health of such soils indicate that they are SOC rich and have less Al-saturation in surface horizons due to the downward movement of Al as organo-metal complexes or chelates, but have higher base saturation than the subsurface horizons (Pal et al., 2014). Despite acidity many such soils show dominance of Ca 2+ and Mg2+ ions on soil exchange complex due to addition of alkaline and alkaline metal cations through litter fall (Nayak et al., 1996; Reza et al., 2018). Such OC-rich acid soils are not kaolinitic as they are dominated by kaolin mineral (a mixed mineral) as their clay CEC is > 24 cmol (p+) kg-1 as determined by BaCl2-TEA for total acidity plus bases by NH4OAc, pH 7 method, (Smith, 1986) and do respond to management interventions that are being made in various land use plans. In view of their excellent support for food production for centuries in the Indian sub-continent and elsewhere (Velayutham and Pal, 2016) it would be wise to

dispel the myth that the lowering of pH or soil acidification is a sign of soil degradation. Finally, it is realized that the above three major pedological and edaphological issues of SAT and HT soils of the tropical world are worthy of consideration in finalising the role of humankind as the 6th soil forming factor and agrogenic degradation, and also to develop the universally acceptable model on agropedogenesis. References Abrol, I. P.: Reclamation and management of salt-affected soils, in Review of soil research in India. Part 11, 12th International Congress Soil Science, New Delhi, pp. 635-654, 1982b. Abrol, I. P.: Reclamation of waste lands and world food prospects, in Whither soil research, Panel Discussion Papers, 12th International Congress Soil Science, New Delhi, pp. 317-337, 1982a. Baulcombe, D., Crute, I., Davies, B., Dunwell, J., Gale, M., Jones, J., Pretty, J., Sutherland, W., Toulmin, C.: Reaping the Benefits: Science and the Sustainable Intensification of Global Agriculture, The Royal Society, pp72. 2009. Bhattacharyya, T., Chandran, P., Ray, S. K., Pal, D. K., Venugopalan, M. V., Mandal, C., Wani, S. P.: Changes in levels of carbon in soils over years of two important food production zones of India, Curr. Sci., 93, 1854–1863, 2007. Bhattacharyya, T., Pal, D. K., Chandran, P., Ray, S. K., Mandal, C., Telpande, B.: Soil carbon storage capacity as a tool to prioritise areas for carbon sequestration, Curr. Sci., 95, 482-494, 2008. Bhattacharyya, T., Pal, D. K., Lal, S., Chandran, P., Ray, S. K.: Formation and persistence of Mollisols on zeolitic Deccan basalt of humid tropical India, Geoderma, 136, 609–620, 2006. Bhattacharyya, T., Pal, D. K., Srivastava, P.: Role of zeolites in persistence of high altitude ferruginous Alfisols of the Western Ghats, India, Geoderma, 90, 263–276, 1999. Bhattacharyya, T., Sarkar, D., Ray, S. K., Chandran, P., Pal, D. K. et al.: Geo referenced soil information system: assessment of database, Curr. Sci., 107, 1400-1419, 2014. Buurman, P., Van Lagen, B., Velthorst, E. J.: Manual of soil and water analysis, Backhuys Publishers, Leiden, 1996. Chandran, P., Ray, S. K., Bhattacharyya, T., Dubey, P. N., Pal, D. K., Krishnan, P.: Chemical and mineralogical characteristics of ferruginous soils of Goa, Clay Res., 23, 51–64, 2004. Chandran, P., Ray, S. K., Bhattacharyya, T., Srivastava, P., Krishnan, P., Pal, D. K.: Lateritic soils of Kerala, India: their mineralogy, genesis and taxonomy, Aust. J. Soil Res., 43, 839–852, 2005.

Chandran, P., Tiwary, P., Mandal, C., Prasad, J., Ray, S. K., Sarkar, D., Pal, D. K. et al.: Development of soil and terrain digital database for major food-growing regions of India for resource planning, Curr. Sci., 107, 1420-1430, 2014. Christensen, B. T., Rasmussen, J., Eriksen, J., Hansen, E. M.: Soil carbon storage and yields of spring barley following grass leys of different age, Eur. J. Agron., 31, 29-35, 2009. Datta, A., Basak, N., Chaudhari, S.K., Sharma, D.K.: Soil properties and organic carbon distribution under different land use in reclaimed sodic soils of North-West India, Geoderma Reg., 4, 134–146, 2015. Datta, A., Mandal, B., Badole, S., Chaitanya, K. A., Mazumdar, S. P., Padhan, D., Basak, N., Barman, A., Kundu, R., Narkhede, W. N.: Interrelationship of biomass yield, carbon input, aggregation, carbon pools and its sequestration in Vertisols under long-term sorghum-wheat cropping system in semi-arid tropics, Soil Tillage Res., 184, 164-175, 2018. Datta, A., Mandal, B., Basak, N., Badole, S., Krishna Chaitanya, A., Majumder, S.P., Thakur, N.P., Kumar, P., Kachroo, D.: Soil carbon pools under long-term ricewheat cropping system in Inceptisols of Indian Himalayas, Arch. Agron. Soil Sci., https://doi.org/10.1080/03650340.2017.1419196, 2017. FAO.: Global agriculture to 2050: how will the world's food and agriculture sector develop in a dynamically changing economic and resource environment? In: How to Feed the World in 2050, FAO, Rome, 2009. Godfray, H. C., Beddington, J. R., Crute, I. R., Haddad, L., Lawrence, D., Muir, J. F, Pretty, J., Robinson, S., Thomas, S. M., Toulmin, C.: Food security: the challenge of feeding 9 billion people, Science, 327, 812–818, 2010. ICAR-NAAS (Indian Council of Agricultural Research- National Academy of Agricultural Sciences).: Degraded and waste lands of India-status and spatial distribution. ICAR-NAAS. Published by the Indian Council of Agricultural Research, New Delhi, pp 56, 2010. Jones, R. C., Hundall, W. H., Sakai, W. S.: Some highly weathered soils of Puerto Rico, 3. Mineralogy, Geoderma, 27, 75–137, doi: 10.1016/0016-7061(82)90048-9, 1982. Koch, A., McBratney, A., Adams, M., Field, D., Hill, R., Crawford, J., Minasny, B., Lal, R., Abbott, L., O'Donnell, A., Angers, D., Baldock, J., Barbier, E., Binkley, D., Parton, W., Wall, D. H., Bird, M., Bouma, J., Chenu, C., Flora, C. B., Goulding, K., Grunwald, S., Hempel, J., Jastrow, J., Lehmann, J., Lorenz, K., Morgan,

C. L., Rice, C. W., Whitehead, D., Young, I., Zimmermann, M.: Soil security: solving the global soil crisis. Glob. Policy, 4, 434–441, 2013. Krishna, C. A., Majumder, S.P., Padhan, D., Badole, S., Datta,A., Mandal, B., and Gade, K.R.: Carbon dynamics, potential and cost of carbon sequestration in double rice cropping system in semi-arid southern India, J. Soil Sci. Plant Nutri., 18 (2), 418-434, 2018. Kuzyakov, Y., Zamanian, K.: Agropedogenesis: humankind 6th soil forming factor and attractors of agrogenic soil degradation. Biogeosciences Discuss., https://doi.org/10.5194/bg-2019-151, 2019. Lal, R.: Sequestering carbon in soils of agro-ecosystems, Food Policy, 36, 33–39, 2011. Lal, R.: Soils and world food security, Soil Tillage Res., 102, 1–4, 2009. Lin, H.: Three principles of soil change and pedogenesis in time and space, Soil Sci. Soc. Amer. J., 75, 2049–2070, 2011. Mandal, B., Majumder, B., Adhya, T.K., Bandyopadhyay, P.K., Gangopadhyay, A., Sarkar, D., Kundu, M.C., Gupta Choudhury, S., Hazra, G.C., Kundu, S., Samantaray, R.N., Mishra, A.K.: Potential of double cropped rice ecology to conserve organic carbon under subtropical climate, Glob. Chang. Biol., 14, 2051–2139, 2008. Mandal, C., Mandal, D. K., Bhattacharyya, T., Sarkar, D., Pal, D. K. et al.: Revisiting agro-ecological sub-regions of India-a case study of two major food production zones, Curr. Sci., 107, 1519- 1536, 2014. Millot, G.: Geology of clays, Springer-Verlag, New York, 1970. Mohr, E. C. J., Van Baren, F. A., van Schuylenborgh, J.: Tropical soils-a comprehensive study of their genesis, Mouton, The Hague, The Netherlands, 1972. Muggler, C. C.: 1998. Polygenetic Oxisols on tertiary surfaces, Minas Gerais, Brazil: soil genesis and landscape development, (PhD thesis) Wageningen Agricultural University, The Netherlands. Nayak, D. C., Sen, T. K., Chamuah, G. S., Sehgal, J. L.: Nature of soil acidity in some soils of Manipur. J. Indian Soc. Soil Sci., 44, 209–214, 1996. Pal, D. K., Bhattacharyya, T., Chandran, P., Ray, S. K.: Tectonics-climate-linked natural soil degradation and its impact in rainfed agriculture: Indian experience, in Wani, S. P., Rockström, J., Oweis, T. (eds) Rainfed agriculture: unlocking the potential. CABI International, Oxfordshire, U.K., pp. 54–72, 2009a. Pal, D. K., Bhattacharyya, T., Ray, S. K., Chandran, P., Srivastava, P., Durge, S. L., Bhuse, S. R.: Significance of soil modifiers (Ca-zeolites and gypsum) in naturally degraded

Vertisols of the Peninsular India in redefining the sodic soils, Geoderma, 136, 210–228, 2006. Pal, D. K., Bhattacharyya, T., Sahrawat, K. L., Wani, S. P.: Natural chemical degradation of soils in the Indian semi-arid tropics and remedial measures, Curr. Sci., 110, 1675- 1682, 2016. Pal, D. K., Bhattacharyya, T., Srivastava, P., Chandran, P., Ray, S. K.: Soils of the Indo-Gangetic Plains: their historical perspective and management, Curr. Sci., 9, 1193-1201, 2009b. Pal, D. K., Bhattacharyya, T., Wani, S. P.: Formation and management of cracking clay soils (Vertisols) to enhance crop productivity: Indian experience, in Lal, R., Stewart, B. A. (eds) World soil resources. Francis and Taylor, Boca Raton, Florida, pp. 317–343, 2012a. Pal, D. K., Dasog, G. S., Vadivelu, S., Ahuja, R. L., Bhattacharyya, T.: Secondary calcium carbonate in soils of arid and semi-arid regions of India, in Lal, R. et al. (eds) Global climate change and pedogenic carbonates, Lewis Publishers, FL, USA, pp.149-185, 2000. Pal, D. K., Sarkar, D., Bhattacharyya, T., Datta, S. C., Chandran, P., Ray, S. K.: Impact of climate change in soils of semi-arid tropics (SAT), in Bhattacharyya et al. (eds) Climate change and agriculture, Studium Press, New Delhi, pp.113-121, 2013. Pal, D. K., Wani, S. P., Sahrawat, K. L., Srivastava, P.: Red ferruginous soils of tropical Indian environments: a review of the pedogenic processes and its implications for edaphology, Catena, 121, 260-278, doi: 10.1016 /j. catena 2014.05.023, 2014. Pal, D. K., Wani, S. P., Sahrawat, K. L.: Carbon sequestration in Indian soils: present status and the potential, Proc. Natl. Acad. Sci., Biol. Sci., India, 85, 337-358, doi:10.1007/s40011-014-0351-6, 2015. Pal, D. K., Wani, S. P., Sahrawat, K. L.: Role of calcium carbonate minerals in improving sustainability of degraded cracking clay soils (Sodic Haplusterts) by improved management: an appraisal of results from the semi-arid zones of India, Clay Res., 31: 94–108, 2012b. Pal, D. K.: A treatise of Indian and tropical soils. Springer International Publishing AG, Cham, Switzerland, 2017. Pathak, H., Byjesh, K., Chakrabarti, B., Aggarwal, P. K.: Potential and cost of carbon sequestration in Indian agriculture: estimates from long-term field experiments, Field Crops Res., 120, 102–111, 2011. Power, A. G.: Ecosystem services and agriculture: trade-offs and synergies, Philos. Trans. R. Soc. Lond. Ser. B Biol. Sci., 365, 2959–2971, 2010. Reza, S. K., Baruah, U., Nayak, D.

C., Dutta, D., Singh, S. K.: Effect of land use on soil physical, chemical and microbial properties in humid subtropical north-eastern India, Natl. Acad. Sci. Lett., doi: https://doi.org/10.1007/s40009-018-0634-1, 2018. Sahrawat, K. L., Bhattacharyya, T., Wani, S. P., Chandran, P., Ray, S. K., Pal, D. K., Padmaja, K. V.: Long-term lowland rice and arable cropping effects on carbon and nitrogen status of some semi-arid tropical soils, Curr. Sci., 89, 2159–2163, 2005. Sandhu, H., Wratten, S., Costanza, R., Pretty, J., Porter, J. R., Reganold, J.: Significance and value of non-traded ecosystem services on farmland, Peer J., 3, e762, 2015. Sehgal, J. L.: Red and lateritic soils: an overview, in Sehgal, J., Blum, W. E., Gajbhiye, K. S. (eds) Red and lateritic soils, Managing red and lateritic soils for sustainable agriculture 1. Oxford and IBH Publishing Co. Pvt. Ltd., New Delhi, pp. 3–10, 1998. Smeck, N. E., Runge, E. C. A., Mackintosh, E. E.: Dynamics and genetic modelling of soil system, in Wilding, L. P., Smeck, N. E., Hall, G. F. (eds) Pedogenesis and soil taxonomy-concepts and interactions, Elsevier, Amsterdam, Developments in Soil Science II-A., pp. 51-81, 1983. Smith, G. D.: The Guy Smith interviews: rationale for concept in Soil Taxonomy, SMSS Technical Monograph 11, SMSS, SCS, USDA, USA, 1986. Tilman, D., Cassman, K. G., Matson, P. A., Naylor, R., Polasky, S.: Agricultural sustainability and intensive production practices, Nature, 418, 671–677, 2002. Tilman, D., Fargione, J., Wolff, B., D'Antonio, C., Dobson, A., Howarth, R., Schindler, D., Schlesinger, W. H., Simberloff, D., Swackhamer, D.: Forecasting agriculturally driven global environmental change, Science, 292, 281–284, 2001. Tiwary, P., Patil, N. G., Bhattacharyya, T., Chandran, P., Ray, S. K., Karthikeyan, K., Sarkar, D., Pal, D. K. et al.: Pedotransfer functions: a tool for estimating hydraulic properties of two major soil types of India, Curr. Sci., 107, 1431-1439, 2014. Varghese, T., Byju, G.: Laterite soils, Technical monograph No.1, State Committee on Science, Technology and Environment, Government of Kerala, Kerala, India, 1993. Velayutham, M. V., Pal, D. K.: Soil Resilience and sustainability of semi-arid and humid tropical soils of India: a commentary, Agropedology, 26, 1-9, 2016. Wani, S. P., Pathak, P., Jangawad, L. S., Eswaran, H., Singh, P.: Improved management of Vertisols in the semi-arid tropics for increased productivity and soil carbon sequestration, Soil Use

Manage., 19

---

## Author Comment (AC1) · 19 Jul 2019

The comments by Dr. Pal focus on agropedogenesis in the tropics. He emphasized that soil development in the tropics under agricultural practices needs particular attention if we aim at developing a universal concept for agropedogenesis. This argument has been considered in section "2.7. Changes in the attractors by specific land-use or climatic conditions" as the agropedogenesis may stop at some metastable conditions depending on specific land-uses or climatic conditions. Nonetheless, agropedogenesis is a universal process leading to similarities in properties of agricultural soils independent on the climatic conditions or other soil forming factors. This is due to the fact that

human activities dominate over the effects of other soil forming factors. The responses to the three major concerns of Dr. Pal are as follow:

(a) The accumulation of soil organic carbon (SOC) by growing agricultural crops in soils of semi-arid tropical (SAT) climate of India which shows no sign of soil degradation.

→ This is true that agricultural practices may also lead to soil improvement. In Fig. 1 we showed both directions i.e. degradation and improvement in soil condition following cultivation onset. In the text also the degradation is mentioned as the most common (but not always) fate of agricultural soils. However, the main message of our paper was the necessity to recognize human as a soil forming factor. Agricultural managements aim at increasing the yield and human, via management practices, modifies the soil properties in the way which it suits crop growth. This makes human as the main factor who determines the direction of changes in soil properties/developments. In consequence, we hypothesized a steady-state condition for soil development under agricultural practices as it is the case in natural pedogenesis. We defined end-values/attractors for a set of properties i.e. master properties which are most sensitive to land-use/land management. These attractors can be as indicators of achieving steady-state condition under agricultural practices. The Dr. Pal's example on organic carbon (OC) of the Indian SAT soils is well in accordance with our attractor concept. As he mentioned the OC content reached to a plateau and a value about 0.7% after nearly three decades of cultivation on Indian SAT soils. Nonetheless, we will try to emphasize more on soil improvement in the revised version of the manuscript to avoid misunderstanding that agricultural practices may solely lead to soil degradation.

(b) The resilience of SAT sodic soils through anthropogenic activities showing pedogenic processes that are reverse to what was proposed in the conceptual model of agropedogenesis.

→ In Fig. 6 we preliminary proposed/defined the time needed for various soil properties to reach their attractors. The attractor of $CaCO_3$ is defined as 0% i.e. complete

decalcification of soil which takes place over millennial time spans. The improved management (IM) system mentioned by Dr. Pal is applying synthetic nitrogen and phosphorus fertilizers as well as furrow irrigation in contrast to the traditional management (TM) system. He argued that implying IM system in SAT sodic soils improved soil condition for crop growth and subsequently increased the yield. However, this conclusion does not rescind our agropedogenesis concept but rather supports it. The modification of alkalinity in SAT sodic soils after implying IM system was due to one order of magnitude increase in $CaCO_3$ solubility. Such an increase in $CaCO_3$ solubility was enough to provide $Ca^{2+}$ ions needed to replace exchangeable $Na^+$ (Pal et al., 2012). This process first confirms the determinant role of human on direction of soil development i.e. agropedogenesis. Second, continuous dissolution of $CaCO_3$ leads to decalcification of the top-soil i.e. movement toward the attractor of $CaCO_3$ (0%) although over millennial period.

(c) Compete with the idea that acidification under agricultural crops is a sign of soil degradation in model concept of agropedogenesis.

$\rightarrow$ We agree that acid soils under for example forest vegetation can still have high organic carbon content. However, in agricultural soils, the yields decrease by decreasing pH value. Thus acidification should be considered as a sign of degradation for agricultural soils.

---

## Short Comment (SC2) · 2 Aug 2019

Further comments by Dr. D.K. Pal and Dr. Ashim Datta on the following responses by the authors Interactive comment on "Agropedogenesis: Humankind as the 6th soil-forming factor and attractors of agrogenic soil degradation" by Yakov Kuzyakov and Kazem Zamanian. The comments by Dr. Pal focus on agropedogenesis in the tropics. He emphasized that soil development in the tropics under agricultural practices needs particular attention if we aim at developing a universal concept for agropedogenesis. This argument has been considered in section "2.7. Changes in the attractors by specific land-use or climatic conditions" as the agropedogenesis may stop at

some metastable conditions depending on specific land-uses or climatic conditions. Nonetheless, agropedogenesis is a universal process leading to similarities in properties of agricultural soils independent on the climatic conditions or other soil forming factors. This is due to the fact that human activities dominate over the effects of other soil forming factors. The responses to the three major concerns of Dr. Pal are as follow: (a) The accumulation of soil organic carbon (SOC) by growing agricultural crops in soils of semi-arid tropical (SAT) climate of India which shows no sign of soil degradation. → This is true that agricultural practices may also lead to soil improvement. In Fig. 1 we showed both directions i.e. degradation and improvement in soil condition following cultivation onset. In the text also the degradation is mentioned as the most common (but not always) fate of agricultural soils. However, the main message of our paper was the necessity to recognize human as a soil forming factor. Agricultural managements aim at increasing the yield and human, via management practices, modifies the soil properties in the way which it suits crop growth. This makes human as the main factor who determines the direction of changes in soil properties/developments. In consequence, we hypothesized a steady-state condition for soil development under agricultural practices as it is the case in natural pedogenesis. We defined end-values/attractors for a set of properties i.e. master properties which are most sensitive to land-use/land management. These attractors can be as indicators of achieving steady-state condition under agricultural practices. The Dr. Pal's example on organic carbon (OC) of the Indian SAT soils is well in accordance with our attractor concept. As he mentioned the OC content reached to a plateau and a value about 0.7% after nearly three decades of cultivation on Indian SAT soils. Nonetheless, we will try to emphasize more on soil improvement in the revised version of the manuscript to avoid misunderstanding that agricultural practices may solely lead to soil degradation.

Response by DKP and AD: Authors' consideration to lay more emphasis on soil development in the revised version of the MS is very much welcome.

(b) The resilience of SAT sodic soils through anthropogenic activities showing pedogenic processes that are reverse to what was proposed in the conceptual model of agropedogenesis. → In Fig. 6 we preliminary proposed/defined the time needed for various soil properties to reach their attractors. The attractor of CaCO3 is defined as 0% i.e. complete decalcification of soil which takes place over millennial time spans. The improved management (IM) system mentioned by Dr. Pal is applying synthetic nitrogen and phosphorus fertilizers as well as furrow irrigation in contrast to the traditional management (TM) system. He argued that implying IM system in SAT sodic soils improved soil condition for crop growth and subsequently increased the yield. However, this conclusion does not rescind our agropedogenesis concept but rather supports it. The modification of alkalinity in SAT sodic soils after implying IM system was due to one order of magnitude increase in CaCO3 solubility. Such an increase in CaCO3 solubility was enough to provide Ca2+ ions needed to replace exchangeable Na+ (Pal et al., 2012). This process first confirms the determinant role of human on direction of soil development i.e. agropedogenesis. Second, continuous dissolution of CaCO3 leads to decalcification of the top-soil i.e. movement toward the attractor of CaCO3 (0%) although over millennial period.

Response by DKP and AD: It is true that the SAT sodic soils will be devoid of CaCO3 over a millennial time scale as the authors envisage when the decalcification process is set in one direction. However, in SAT environment the formation of pedogenic CaCO3 (PC), though at a much slower rate than its dissolution, will continue to provide fresh stock of PC (Pal et al., 2000). Under such chemical environment it will be difficult to presume that SAT soils under agricultural crops would ever become non-calcareous. Therefore, authors are urged to consider this type of soil development in their conceptual model on agropedogenesis, which is inclined more towards soil degradation.

(c) Compete with the idea that acidification under agricultural crops is a sign of soil degradation in model concept of agropedogenesis. → We agree that acid soils under for example forest vegetation can still have high organic carbon content. However, in agricultural soils, the yields decrease by decreasing pH value. Thus acidification
should be considered as a sign of degradation for agricultural soils. Response by DKP and AD: Once again it is reminded that tropical acid soils were considered to be chemically degraded because of their high acidity (pH «6.5), caused by the profuse chemical weathering under humid tropical climate. They were often conceived to be soils that have less soil fertility generally. This contention strongly contrasts with their OC enrichment (> 1%) in the surface horizons even in agricultural soils (Pal et al., 2014,Pal 2019). The present health of such OC rich soils has less Al-saturation in surface horizons due to the downward movement of Al as organo-metal complexes or chelates, but has higher base saturation than the subsurface horizons. Despite acidity many such soils show dominance of Ca 2+ and Mg2+ ions on soil exchange complex due to addition of alkaline and alkaline metal cations through litter fall (Nayak et al. 1996; Reza et al. 2018).Such OC-rich acid soils are not kaolinitic as they are dominated by kaolin mineral (a mixed mineral) and do respond to management interventions that are being made in various land use plans. Moreover, in such million years old Alfisols, Mollisols and Ultisols of the Indian sub-continent are still continuing to provide ecosystem services in supporting agriculture, horticulture, forestry, tea, coffee and spices (Pal et al., 2014, Pal, 2017). Referring to unique Indian experience, it would be wise to dispel the myth that acid soils under agricultural land use plans in accordance to the national agricultural research system, remarkably lowers the soil pH or causes further soil acidification to the extent that these soils stop support for plants to grow. Finally, the acid soils under national agricultural research system are not being degraded by the current agropedogenesis. Therefore, further attention of the authors is once again being sought to reconsider the concept that acidification is a sign of soil degradation. References Nayak, D. C., Sen, T. K., Chamuah, G. S., Sehgal, J. L.: Nature of soil acidity in some soils of Manipur. J. Indian Soc. Soil Sci., 44, 209–214, 1996. Pal, D. K.: A treatise of Indian and tropical soils. Springer International Publishing AG, Cham, Switzerland, 2017. Pal, D. K.: Simple methods to study pedology and edaphology of Indian tropical soils. Springer International Publishing AG, Cham, Switzerland, 2019. Pal, D. K., Dasog, G. S., Vadivelu, S., Ahuja, R. L., Bhattacharyya, T.: Secondary calcium carbonate in soils of arid and semi-arid regions of India. In: Lal R. et al. (eds) Global climate change and pedogenic carbonates, Lewis Publishers, FL, USA, pp149-185, 2000. Pal, D. K., Wani, S. P., Sahrawat, K. L., Srivastava, P.: Red ferruginous soils of tropical Indian environments: a review of the pedogenic processes and its implications for edaphology. Catena 121, 260-278, 2014. doi:10.1016 /j. catena 2014.05.023 Reza, S. K., Baruah, U., Nayak, D. C., Dutta, D., Singh, S. K.: Effect of land use on soil physical, chemical and microbial properties in humid subtropical north-eastern India. Natl. Acad. Sci. Lett., 2018. https://doi.org/10.1007/s40009-018-0634-1

---

## Referee Comment (RC1) · Anonymous Referee #1 · 3 Sep 2019

This review paper addresses humankind impacts on soil development. The authors highlight the importance of humankind impact as new soil formation factor and distinguish it from natural soil formation factor due to the impact that it has on the soil development. As the authors pointed out in their text the importance of humankind impacts on soil formation has been acknowledged by some researchers but what makes the view of authors special here is the way they take into account its contribution in soil development. They argue that the natural soil processes result in soils with diverse functions and properties, while the humankind interferences in the ecosystem result in soils with uniform and similar functions and properties. In this sense, the impact of humankind on soil development is introduced as a convergence factor and neutral soil

formation factors as a divergence factor.

The authors' opinion here is mainly supported by some examples at which different land uses (mainly forest) were converted to agricultural use. I found the view of authors interesting and considered it as an emerging topic in the field of fundamental soil science. In general, I do not have any fundamental comments on the concept presented here and believe that this review should be published as a review paper in the journal of Biogeosciences Discussion.

Given that all the authors are very experienced scientists with a substantial track record, this is a pity, and I cannot refrain from emphasizing that the text and figures need some careful revisions. Some examples are listed below:

Fig.1 is an interesting figure showing the main concept presented in this review. However, it was hard for me to follow its context and would suggest some modifications to this figure as follows: 2) place the legend on the right side of the figure. In its current location is confusing and the readers may relate it to the time, 2) Does the red arrow on x-axis show start of cultivation decades? if yes remove its label out of the figure that one can read it. otherwise, it looks like two different labelings,3) it is not clear what does it show the label " duration/intensity of cultivation. Do you mean a time period between the start of cultivation till now? If yes, show it with an arrows bellow the x-axis, 4) move the label of x-axis more to the bottom and make some space with indicated time.

In fig. 2, what does it mean 'Soil genesis based on the development of concepts' in the caption of figure? I would recommend the authors to rearrange this figure and improve its readability. In the current version, it is hard to follow its context and massage. Found a better away of relating this information together, for instance, the factors and parental materials, climate, etc. Here and elsewhere in the figures, I found it annoying for readers to follow a diagram with varying font sizes and styles.

In Fig. 4: It is hard to understand the message of this figure. What does it mean factors

2: 38% and 1:48% in the label of x-axis and y-axis. Do you mean a relative increase of 38% and 48%? Where does the 1 start?

Fig. 5: rephrase the caption, it is a confusing sentence and hard to read. In Fig. 5a and 5b, explain in the legend what do show the solid lines. The legend of Fig. 5c and 5d are confusing. Use a separate legend for every four cases.

Fig. 6: This is an interesting figure. State that this is a hypothetical trend. How do the authors argue on the proposed time? It looked to me that the authors aimed to show here the relative responses of each process with time and the selection of time is not based on any experimental evidence. If that is true I suggest using a normalized time between 0 and 1 to avoid giving a weak impression.

Fig. 9: how did the authors generate these figures? Are they hypothetical figures? If yes mention it in the caption. What does it mean stage in these figures? Stage of what?

Some minor typos:

Line 220: Replace "decreases " with "decrease "

Line 33: replace 'fulfils' with 'fulfills'

Line 378: replace because with become

Line 279: replace "independent of" with "independency of"

Line 149: Do the author mean the function rather than production?

Line 138: Replace " develops" with " develop."
* * *

---

## Short Comment (SC3) · 4 Sep 2019

The authors introduce a theory of anthropedogenesis – soil development under the main factor 'humankind' – the 6th factor of soil formation, and deepen it to encompass agropedogenesis as the most important direction of anthropedogenesis. The theory of agropedogenesis is a very important issue in pedology and there is a clear gap in knowledge related to this issue and the outcomes of this research certainly help to better understand the dynamics of soil development under agricultural practices. Although the contents of the manuscript are fairly good, it would benefit from better editing (e.g. grammar and clarity), which would improve its clarity. In addition, some

necessary improvements are suggested in the following:

1) It is also important to discuss more thoroughly, why these soil properties were selected [Master soil properties]. In particular, a reader would like to know whether these soil properties are intrinsically more important than the others or simply more important in this study due to some identified characteristics and assumptions. 2) It is necessary to explain clearly the figures in the main body of the manuscript.

Some other comments are made along with the text: Keywords: I think five keywords are enough. Line 4-5: This first sentence of the abstract should be removed. Line 48-49: Please clarify this sentence "Since the suitable land resources for agriculture are limited and increasingly located in ecologically marginal conditions". Line 50: add cit. Line 73: run-off irrigation and terracing Line 80: add cit. Line 87: "The human factor can even change soil types as defined by classification systems (Supplementary Fig. 1)". Figure 1 indicates the convergence and divergence of soil properties! Line 104: add cit. Table 2: justify Table 2 Line 122: climate, organisms, relief and time Line 139: climate, organisms, and relief Line 140: "...over time. Thus, morphological soil properties...". This sentence should be rewritten. Line 143: Figure 2. Line 153: add cit. Line 180: climate, organisms, and relief Line 201: How is possible to infer the decreasing in the spatial variability of soil properties in figure 5. Line 847: "(c) and (d) total soil carbon"! Lines 273-lines 299: the definition of phase diagrams would be necessary. Not sure that every Biogeosciences reader is familiar with them.

Please also note the supplement to this comment:
https://www.biogeosciences-discuss.net/bg-2019-151/bg-2019-151-SC3-supplement.pdf

---

## Referee Comment (RC2) · Anonymous Referee #2 · 6 Sep 2019

The authors introduce a theory of anthropedogenesis – soil development under the main factor 'humankind' – the 6th factor of soil formation, and deepen it to encompass agropedogenesis as the most important direction of anthropedogenesis. The theory of agropedogenesis is a very important issue in pedology and there is a clear gap in knowledge related to this issue and the outcomes of this research certainly help to better understand the dynamics of soil development under agricultural practices. Although the contents of the manuscript is fairly good, it would benefit from better editing (e.g. grammar and clarity), which would improve its clarity. In addition, some necessary improvements are suggested in the following:

1) More comprehensive literature review on soils [e.g. semi-arid tropical soils] showing no sign of soil degradation by growing agricultural crops in soils. 2) It is also important to discuss more thoroughly, why these soil properties were selected [Master soil properties]. In particular, a reader would like to know whether these soil properties are intrinsically more important than the others or simply more important in this study due to some identified characteristics and assumptions. 3) It is necessary to explain clearly the figures in the main body of the manuscript.

Some other comments are made along with the text: Keywords: I think five keywords are enough. Line 4-5: This first sentence of the abstract should be removed. Line 48-49: Please clarify this sentence "Since the suitable land resources for agriculture are limited and increasingly located in ecologically marginal conditions". Line 50: add cit. Line 73: run-off irrigation and terracing Line 80: add cit. Line 87: "The human factor can even change soil types as defined by classification systems (Supplementary Fig. 1)". Figure 1 indicates the convergence and divergence of soil properties! Line 104: add cit. Table 2: justify Table 2 Line 122: climate, organisms, relief and time Line 139: climate, organisms, and relief Line 140: "...over time. Thus, morphological soil properties...". This sentence should be rewritten. Line 143: Figure 2. Line 153: add cit. Line 180: climate, organisms, and relief Line 201: How is possible to infer the decreasing in the spatial variability of soil properties in figure 5. Line 847: "(c) and (d) total soil carbon"! Lines 273-lines 299: the definition of phase diagrams would be necessary. Not sure that every Biogeosciences reader is familiar with them.

---

## Short Comment (SC4) · 6 Sep 2019

It was a pleasure to read the manuscript. I have some minor remarks, which may improve the strength of the discussion, if considered. Best wishes, Peter Kühn

General Remarks Chapters 1.2 and 2.1 In this context the scorpan model by McBratney et al. (2003; "On digital soil mapping") should be discussed as well, which includes more than five soil forming factors and particularly their functions.

188-190: If the "convergence of soil properties" is not true in all cases, I recommend rephrasing the statement in line 188.

[Figure]

Chapter 2.7 Additionally different topographic positions should be discussed: upslope, midslope, toeslope and even positions. Do not soil properties diverge or converge despite of human impact just related to the topographic position of the soil? E.g. imagine calcareous substrate with a decalcified soil, at upslope positions and human-induced soil erosion; after some time the soil will have many properties of the substrate, particularly regarding carbonate content, pH, EC, and the content of some elements as e.g. Ca and Mg. These are also master properties of agropedogenesis as you defined in chapter 2.4. - And e.g. in toeslope positions you have often an additional material input from upslope positions, which influences also some master properties and might rule out convergent tendecies. Of course this is different under humid and arid climate conditions.

---

## Author Response (AR1)

**Response to the Reviewers' comments**

**Anonymous Referee #1**

This review paper addresses humankind impacts on soil development. The authors highlight the importance of humankind impact as new soil formation factor and distinguish it from natural soil formation factor due to the impact that it has on the soil development. As the authors pointed out in their text the importance of humankind impacts on soil formation has been acknowledged by some researchers but what makes the view of authors special here is the way they take into account its contribution in soil development. They argue that the natural soil processes result in soils with diverse functions and properties, while the humankind interferences in the ecosystem result in soils with uniform and similar functions and properties. In this sense, the impact of humankind on soil development is introduced as a convergence factor and neutral soil formation factors as a divergence factor.

The authors' opinion here is mainly supported by some examples at which different land uses (mainly forest) were converted to agricultural use. I found the view of authors interesting and considered it as an emerging topic in the field of fundamental soil science. In general, I do not have any fundamental comments on the concept presented here and believe that this review should be published as a review paper in the journal of Biogeosciences Discussion.

We are very thankful to the Reviewer for his very positive assessment and suggested improvements.
Please see our improvements and answers below.

Given that all the authors are very experienced scientists with a substantial track record, this is a pity, and I cannot refrain from emphasizing that the text and figures need some careful revisions. Some examples are listed below:

Fig.1 is an interesting figure showing the main concept presented in this review. However, it was hard for me to follow its context and would suggest some modifications to this figure as follows: 2) place the legend on the right side of the figure. In its current location is confusing and the readers may relate it to the time, 2) Does the red arrow on x-axis show start of cultivation decades? if yes remove its label out of the figure that one can read it. otherwise, it looks like two different labelings,3) it is not clear what does it show the label " duration/intensity of cultivation. Do you mean a time period between the start of cultivation till now? If yes, show it with an arrows bellow the x-axis, 4) move the label of x-axis more to the bottom and make some space with indicated time.

Many thanks – we can understand well that these points are not clear.
We improved the Fig 1 as suggested by the Reviewer and hope that it is easier to follow now.
We added legend, removed Millenia and Decades, added additional x axis for agropedogenesis.

In fig. 2, what does it mean 'Soil genesis based on the development of concepts' in the caption of figure? I would recommend the authors to rearrange this figure and improve its readability. In the current version, it is hard to follow its context and massage. Found a better away of relating this information together, for instance, the factors and parental materials, climate, etc. Here and elsewhere in the figures, I found it annoying for readers to follow a diagram with varying font sizes and styles.

We completely rewrote the caption.
We have unified better the font sizes within each Fig. We still left some various fonts to show the importance of processes.

In Fig. 4: It is hard to understand the message of this figure. What does it mean factors 2: 38% and 1: 48% in the label of x-axis and y-axis. Do you mean a relative increase of 38% and 48%? Where does the 1 start?

This is the results of a principal component analysis on various parameters measured in the abandoned agricultural soils with increasing abandonment periods. We improved the Figure and also add more details to the legend for better understanding. 75% of variation in soil properties is explained by factor 1 and 19% by factor 2.
If the Reviewer mean that this is superfluous Fig., we will move it to Supplementary Materials.

Fig. 5: rephrase the caption, it is a confusing sentence and hard to read. In Fig. 5a and 5b, explain in the legend what do show the solid lines. The legend of Fig. 5c and 5d are confusing. Use a separate legend for every four cases.

The fig. caption has been modified.

Fig. 6: This is an interesting figure. State that this is a hypothetical trend. How do the authors argue on the proposed time? It looked to me that the authors aimed to show here the relative responses of each process with time and the selection of time is not based on any experimental evidence. If that is true I suggest using a normalized time between 0 and 1 to avoid giving a weak impression.

The fig. is actually based on the real values stated for each soil property in various studies (including that presented in the Fig. 3 and 4). Nonetheless, the values written on each curve are our suggestion for the attractor of each soil property over long-term cultivation. See also line 380-385.

Fig. 9: how did the authors generate these figures? Are they hypothetical figures? If yes mention it in the caption. What does it mean stage in these figures? Stage of what?

The figures are conceptual phase diagrams as it is mentioned in the caption. These phase diagrams were made based on the curves in the Fig 6 (now 5), which are experimentally based. The stages show the changing trend in a given soil property over the degradation processes.

The stages are time laps to reach a threshold for a given soil property when after that the trend may slow down or become reversed. See line 291-292 for definition of stages of degradation. The fig. caption has been modified.

Some minor typos:
Line 220: Replace "decreases " with "decrease "

Decreases in Line 234 has revised

Line 33: replace 'fulfils' with 'fulfills'

It is revised in Line 35

Line 378: replace because with become

The sentence has been modified

Line 279: replace "independent of" with "independency of"

"Independent of" looks grammatically correct here.

Line 149: Do the author mean the function rather than production?

No, the (crop) production is one of the soil functions. So, when only one function can be increased at a time the other functions (other than production) will be decreased.

Line 138: Replace " develops" with " develop."

The sentence has been modified

……………

**Anonymous Referee #2**

The authors introduce a theory of anthropedogenesis – soil development under the main factor 'humankind' – the 6th factor of soil formation, and deepen it to encompass agropedogenesis as the most important direction of anthropedogenesis. The theory of agropedogenesis is a very important issue in pedology and there is a clear gap in knowledge related to this issue and the outcomes of this research certainly help to better understand the dynamics of soil development under agricultural practices.

We are very thankful for this positive evaluation and suggested improvements.
Please see our improvements and answers below.

Although the contents of the manuscript is fairly good, it would benefit from better editing (e.g. grammar and clarity), which would improve its clarity.

We sent the ms once again for the improvement of the English language.

In addition, some necessary improvements are suggested in the following:

1) More comprehensive literature review on soils [e.g. semi-arid tropical soils] showing no sign of soil degradation by growing agricultural crops in soils.

This point is based on the comments given by Dr. Pal about the necessity to exclude semi-arid tropical soils from the concept of agropedogenesis. The point that Dr. Pal emphasized to be "no-sign-of-degradation" is solely based on stability of SOC content over 25 years of cultivation in semi-arid tropical soils of India. This is however, because of yearly addition of large amount of organic fertilizers which keeps the SOC content at a high level along with the presence of alkaline soils which prohibit soil acidification. This, in our opinion, is temporary condition (i.e. pedogenic inertia) and following decalcification of topsoil (when attractor of CaCO3 is achieved) the mentioned soils will also face acidification and so, degradation and crop reduction. We already addressed in the text that such conditions may also take place (see lines 210-211) due to soil intrinsic master properties which are from their threshold values to cause soil degradation.

2) It is also important to discuss more thoroughly, why these soil properties were selected [Master soil properties]. In particular, a reader would like to know whether these soil properties are intrinsically more important than the others or simply more important in this study due to some identified characteristics and assumptions.

The main characteristic of a soil property to be a master property in agropedogenesis concept is its sensitivity to agricultural use. Further, changes in the values of the so-called master properties should determine the state many other properties over cultivation period. See section 2.4 as we defined the master properties and their particular characteristics. Also the most other studies suggested these properties (see Table 3).
We would like to discuss these soil properties and the reasons in the next paper. This paper is already too long for individual description of each of the nine properties.

3) It is necessary to explain clearly the figures in the main body of the manuscript.

We agree. The Reviewer #1 mentioned the same. In the improved version we presented more explanations and details to the figure legend.

Some other comments are made along with the text:
Keywords: I think five keywords are enough.

We developed a theory which is not only connected to the effects of human on soil conditions but also to the effects of human in general on planet Earth and so, to the Anthropocene. This includes many aspects which we tried to address by the key-words for a better indexing by the searching programs.
We deleted 4 Keywords (but added 2).

Line 4-5: This first sentence of the abstract should be removed.

This sentence actually shows the relevance and significance of studying the effects of human on agricultural soils. It shows that human through agricultural practices may affect a huge land surface area. Deleting this sentence will raise the question of how significant or relevant is this study.
If the Reviewer insists on it, we will delete this sentence.

Line48-49: Please clarify this sentence "Since the suitable land resources for agriculture are limited and increasingly located in ecologically marginal conditions".

The suitable land areas for agricultural practices are limited. Therefore, many studies are focusing on protecting strategies to save such areas against degradation causing decreasing food production. Furthermore, if intensification in crop production on the available land is not considered then, we have to cultivate the ecologically susceptible areas for example shallow soils on steep slopes. We simplified the sentence.

Line 50: add cit.

Lal et al., 2005 has been added.

Line 73: run-off irrigation and terracing

"and" has been added.

Line 80: add cit.

FAO 2018 has been added.

Line 87: "The human factor can even change soil types as defined by classification systems (Supplementary Fig. 1)".

The sentence is correct similar to what the reviewer has written.

Figure 1 indicates the convergence and divergence of soil properties!

Under natural soil genesis, yes (the green lines) but convergence under agropedogenesis (red lines). The fig. is however, improved for better clarifications.

Line104: add cit.

See Dudal, 2004 (line 101).

Table 2: justify Table 2

We wanted to bold the main soil formation processes under agricultural practices and their consequences on soil properties. Could you please let us know what you mean with justifying the table?

Line 122: climate, organisms, relief and time

It has been revised accordingly.

Line 139: climate, organisms, and relief

It has been revised accordingly.

Line 140: "...over time. Thus, morphological soil properties...". This sentence should be rewritten.

The sentence is re-written as: Therefore, visible morphological soil properties in the field and measurable parameters in the lab were very well described leading to development of various (semi)genetic soil classifications

Line 143: Figure 2.

Corrected

Line 153:add cit.

This is authors definition of soil degradation and its stages.

Line 180: climate, organisms, and relief

It has been revised accordingly.

Line 201: How is possible to infer the decreasing in the spatial variability of soil properties in figure 5.

The sentence has been corrected.

Line 847: "(c) and(d) total soil carbon"!

The sentence has been corrected.

Lines 273-lines 299: the definition of phase diagrams would be necessary. Not sure that every Biogeosciences reader is familiar with them.

We added the definition of the phase diagrams (see line 277).

**# Other comments and minor corrections by Peter Kühn**

It was a pleasure to read the manuscript. I have some minor remarks, which may improve the strength of the discussion, if considered. Best wishes, Peter Kühn

We are very thankful to Prof. Kühn for his positive assessment and suggested improvements.

General Remarks Chapters 1.2 and 2.1 In this context the scorpan model by McBratney et al. (2003; "On digital soil mapping") should be discussed as well, which includes more than five soil forming factors and particularly their functions.

The reference McBratney et al. (2003) has been added to the text.

188-190: If the "convergence of soil properties" is not true in all cases, I recommend rephrasing the statement in line 188.

The sentence has been deleted.

Chapter 2.7 Additionally different topographic positions should be discussed: upslope, Mid-slope, toe-slope and even positions. Do not soil properties diverge or converge despite of human impact just related to the topographic position of the soil? E.g. imagine calcareous substrate with a decalcified soil, at upslope positions and human-induced soil erosion; after some time the soil will have many properties of the substrate, particularly regarding carbonate content, pH, EC, and the content of some elements as e.g. Ca and Mg. These are also master properties of agropedogenesis as you defined in chapter 2.4. - And e.g. in toe-slope positions you have often an additional material input from upslope positions, which influences also some master properties and might rule out convergent tendencies. Of course this is different under humid and arid climate conditions.

We assumed that agricultural soils are generally located on flat and leveled grounds or on gentle slopes and there would be terracing on steeper slopes. On the other hand, we hypothesized that there will be an equilibrium between the erosion rate and soil genesis rate over long time farming (see supplementary fig. 1).

Cover page

[revised manuscript text omitted]

⇩ total porosity
⇩ water holding capacity
⇩ soil aeration | - ⇩ root density
- ⇩ burrowing animals
- ⇩ large & medium aggregates | (Celik, 2005; Lipiec et al., 2012)
(Flynn et al., 2009; Ponge et al., 2013) |
| | ⇩ soil depth | - ⇧ water and wind erosion
- ⇧ tillage erosion
- ⇧ soil density | (Ayoubi et al., 2012; Govers et al., 1994; Lal, 2001) |
| **Chemical properties** | ⇩ SOM content
⇩ easily available and low molecular weight organic substances | - ⇧ SOM mineralization by increasing aeration
- removal of plant biomass via harvesting
- residual burning
- destruction of macro-aggregates | (Lisetskii et al., 2015; Liu et al., 2009; Sandor and Homburg, 2017) |
| | ⇩ element/nutrient content
loss of nutrients
narrowing of C:N:P ratio | - removal of plant biomass via harvesting
- nutrient leaching
- SOM mineralization + NP-fertilization | (Hartemink, 2006; Lisetskii et al., 2015; Sandor and Homburg, 2017) |

| | | | |
|---|---|---|---|
| | Acidification:
⇓     pH
⇑     exchangeable aluminum
⇓     CEC | - N-fertilization
- cation removal by harvest
- ⇓ buffering capacity due to cation leaching and decalcification
- acidification and $H^+$ domination on exchange sites
- loss of SOM | (Homburg and Sandor, 2011; Obour et al., 2017; Zamanian and Kuzyakov, 2019) |
| | ⇑ salts and/or exchangeable $Na^+$ | - irrigation (with low-quality water or/and groundwater level rise by irrigation) | (Dehaan and Taylor, 2002; Emdad et al., 2004; Jalali and Ranjbar, 2009; Lal, 2015) |
| Biological properties | ⇓ biodiversity
⇓ (micro)organism density and abundance | - weeding
- pesticide application
- monocultures or narrow crop rotations
- mineral fertilization
- ⇓ SOM content and litter input
- ⇓ root amounts and rhizosphere volume
- plowing and grubbing
- ⇓ total SOM
- pesticide application | (Lal, 2009; Zhang et al., 2017)
(Breland and Eltun, 1999; Fageria, 2012) |
| | ⇓ microbial activities
 - respiration
  - enzyme activities | - recalcitrance of remaining SOM
- ⇓ microbial abundance
- ⇓ litter & rhizodeposition input
- mineral fertilization
- ⇓ organism activity, diversity and abundance
- shift in microbial community structure
- ⇓ soil animal abundance and activity | (Breland and Eltun, 1999) (Bosch-Serra et al., 2014; Diedhiou et al., 2009; Ponge et al., 2013) |

⇑ and ⇓ means increase or decrease, respectively

**Table 2: Soil formation processes under agricultural practices**

| Additions | Losses | Translocation | Transformation |
|---|---|---|---|
| Irrigation
 - water
 - salts ⇧*
 - sediments | Mineralization ⇧
 - organic matter
 - plant residues
 - organic fertilizers
 -  N (to $N_2O$ and $N_2$) ⇧ | Irrigation
 - dissolved organic matter ⇩
 - soluble salts ⇧ | Fertilization
 - acceleration of nutrient (C, N, P, etc.) cycles
 - formation of potassium-rich clay minerals |
| Fertilization:
 - mineral
 - organic (manure, crop residues) | Erosion:
 - fine earth erosion ⇧
 - whole soil material | Evaporation
 - soluble salt transportation to the topsoil ⇧ | Mineralization ⇧
 - humification of organic residues ⇩
 - organo-mineral interactions ⇩ |
| Pest control
 - pesticides
 - herbicides | Leaching:
 - nutrients leaching ⇧
 - cations ⇧
 - $CaCO_3$ | Plowing/deep plowing
 - soil horizon mixing
 - homogenization
 - bioturbation ⇩ | Heavy machinery
 - compaction of top- and subsoil
 - aggregate destruction ⇧ |
| Amendments
 - liming
 - gypsum
 - sand**
 - biochar | Harvesting
 - nutrients
 - ballast (Si, Al, Na, …) elements | | Pest control

[revised manuscript text omitted]

Florinsky, I.V. 2012. The Dokuchaev hypothesis as a basis for predictive digital soil mapping (on the 125$^{th}$ anniversary of its publication). Eurasian Soil Science 45 (4), 445-451.

[Figure]

[Figure]

Supplementary Fig. 1: Soil depth decrease due to erosion. The erosion rate decreases with cultivation duration due to depletion of easily erodible materials. It reaches steady state conditions when erosion becomes equal to soil genesis. After major erosion, the soil taxonomic group changed due to a strong decrease in the Ah / Ap horizon depth, which led to new qualifiers and master properties. Other frequent examples of soil class changes are presented in Dudal (2004)

[Figure]

Supplementary Fig. 2: Examples of convergence of soil properties as a result of cultivation duration: (top) Nitrate content, (bottom) ammonium content depending on soil depth during 20 years of cultivation (Jones and Dalal, 2017). The solid lines are added to better visualize the changing trends in nitrate and ammonium contents as a function of cultivation duration.

Supplementary Fig. 3: Extended conceptual schema of convergence of soil properties by agropedogenesis (see also Fig. 9). The very broad range of natural soils and their properties will be tailored for crop production by agricultural use, resulting in Anthrosols with a very narrow range of properties – the convergence of properties by agropedogenesis. Note that the soils within the funnel are mentioned exemplarily and not all WRB soil groups are presented. The sequence of soils within the funnel does not reflect their transformations during agropedogenesis to Anthrosols. This extended version reflecting multiple pathways to Anthrosols and their variability. Nevertheless, the variability of all soil parameters is much lower compared to natural soils.